# Wind loads and competition for light sculpt trees into self-similar structures

Christophe Eloy [1], Meriem Fournier[2], André Lacointe[3] & Bruno Moulia[3]

Trees are self-similar structures: their branch lengths and diameters vary allometrically within the tree architecture, with longer and thicker branches near the ground. These tree allometries are often attributed to optimisation of hydraulic sap transport and safety against elastic buckling. Here, we show that these allometries also emerge from a model that includes competition for light, wind biomechanics and no hydraulics. We have developed MECHATREE, a numerical model of trees growing and evolving on a virtual island. With this model, we identify the fittest growth strategy when trees compete for light and allocate their photosynthates to grow seeds, create new branches or reinforce existing ones in response to wind-induced loads. Strikingly, we find that selected trees species are self-similar and follow allometric scalings similar to those observed on dicots and conifers. This result suggests that resistance to wind and competition for light play an essential role in determining tree allometries.

[1] Aix Marseille Univ, CNRS, Centrale Marseille, F-13013 IRPHE Marseille, France. [2] LERFoB, INRA, AgroParisTech, F-54000 Nancy, France. [3] UCA, INRA, UMR PIAF, F-63000 Clermont-Ferrand, France. Correspondence and requests for materials should be addressed to C.E. (email: eloy@irphe.univ-mrs.fr)

Tree branching networks are generally self-similar. As a result, the diameters and lengths of branches decrease with the distance to the ground[1]. However, newly grown branch segments always have approximately the same length[2]. The observed hierarchy of lengths is in fact due to complex reconfigurations through the continuous growth of new branches and the pruning of old ones. Self-similarity is thus intimately linked to the growth history of trees.

Self-similar properties of individual trees can be quantified by measuring the radii, lengths and masses of each branch. Such measurements have indicated that these quantities vary allometrically, as expected for self-similar structures[3, 4]. Interestingly, allometric scalings are also observed when comparing inter or intraspecifically different populations of trees[1, 5, 6]. These scalings, which are usually what is meant by 'tree allometry', relate tree height, stem biomass, trunk diameter, etc.

Two classes of theoretical explanations have been given to account for these allometric laws: mechanical[4, 7–9] and hydraulic[10–12]. Mechanical explanations date back to the work of Metzger[13], who proposed that wind-induced stresses should be constant along trunks. This argument is related to the 'axiom of uniform stress'[14, 15], a necessary condition to minimise the amount of material needed to support a load. Optimal mechanical design is associated to the now well-established process of thigmomorphogenesis, which is the plant response to mechanical signals[16, 17]. The ecological significance of thigmomorphogenetic acclimation has long been recognised[18], even if it has yet to be implemented in ecological forest models.

Among the mechanical arguments, the concept of 'elastic similarity' has often been used[4]. Elastic similarity is an allometric law that relates branch radii and lengths, such that the deflection of the branch tip under self-weight is proportional to its length. The same allometric law can be recovered for upright axes when a constant safety factor against elastic buckling is enforced[4]. Although it is generally admitted that wind loads offer a bigger challenge to trees than buckling[19, 20], elastic similarity is the main mechanical component of many allometric models[5, 21, 22].

Within the hydraulic models, the pipe model[10] or the initial version of the West, Brown and Enquist (WBE) model[23] have been highly influential. In these models, a tree is modelled as a fractal assembly of sap-conducting pipes. In its current version however, the WBE model for plants[21], as well as related models[5, 22], also include a mechanical principle: trees are modelled as volume-filling networks following the principle of elastic similarity. With this approach, several allometric laws can be deduced, relating trunk radius, tree height, stem biomass and leaf biomass.

Mechanical and hydraulic models have often been compared, opposed or combined[9, 11, 24], but both rely on simplifications to be questioned here. First, most models do not consider explicitly the evolutionary mechanisms[25]. Second, tree architectures are generally prescribed, without addressing growth. Therefore, they cannot reflect the specific reconfiguration mechanisms found in trees. Considering tree growth is the viewpoint of functional-structural models[26], which consider plants as an assembly of individual organs, explicitly describing development and carbon allocation[27]. These models, such as LIGNUM[28], GREENLAB[29], AMAP[30] or L-PEACH[31] are usually based on a large number of empirical parameters with the aim of modelling particular species. An alternative approach is to exploit the recursive characteristics of tree architectures by using the formal grammar of L-systems[32]. With this approach, the self-organising processes associated with growth, competition for light and interactions with the environment can be addressed[33]. However, in these models, evolutionary processes and wind biomechanics are usually neglected.

To address the limitations of past approaches, we propose MECHATREE, a new functional-structural model of tree growth. This model integrates important biological processes related to growth, architecture reconfiguration and evolution: competition for light, carbon allocation, thigmomorphogenesis, wind-induced pruning and genetic evolution. The main novelty of MECHATREE lies in its ability to compute the growth and evolution of entire ecosystems. We will use this feature to address two important questions: What is the carbon allocation strategy best adapted to competition for light, resistance to wind and reproduction needs? Do the selected growth strategies yield tree architecture and allometry consistent with empirical laws?

In MECHATREE, the modelling units are branch segments. In contrast, models such as SORTIE[34], ITD[35] or SERA[36] are individual-based: the modelling units are individual trees with species-specific allometric parameters. These models can address the dynamics of a forest ecosystem, but are not suited to study the origin of allometric scalings in individual tree architectures.

In the present study, we simulate a simplified version of evolution on uniform virtual islands, with no gene influx at the boundary. This island ecosystem is known to rapidly lead to a single or very few dominant adapted species[37]. To speed up genetic divergence between lines, we further assume autogamy (child and parent share the same genome, except for slight mutations). This island environment is submitted to realistic recurrent wind gales[38], and resource competition is limited to light accessibility. In this simplified environment, in silico evolution allow for a direct falsification of the selective force behind allometric scalings. With this model, we find that self-similarity of branch lengths and fractal dimension emerge through competition for light, while branch diameters are set through the response to wind-induced stresses and eventually yield Leonardo's rule of area conservation.

## Results

**Structural units**. Trees are modelled as modular structures of different units: segments, foliages and seeds, together with a carbon reserve. New units can be added, provided sufficient photosynthates have been produced, and units can be pruned by wind. Our aim is to mimic the main characteristics of an angiosperm-like phenotypic set[39]. In building MECHATREE, several simplifications have been made, with the goal of keeping the model parsimonious and manageable. In particular, we have neglected the selective pressure exerted by hydraulics through the cost of transport and embolism. This is by no means because hydraulics is not important, but we wanted to assess whether a model based on light competition and wind-induced alone could predict realistic allometries.

Tree branches are assemblies of segments, which are cylinders of varying diameter $d$, but with always the same length $L$

**Table 1 Parameters of MECHATREE**

| Parameter | Symbol | Value |
|---|---|---|
| Segment length | $L$ | Arbitrary |
| Twig diameter | $d_0$ | $0.1L$ |
| Twig volume | $V_0$ | $\pi L d_0^2/4$ |
| Foliage diameter | – | $L$ |
| Foliage transparency | $\alpha_{fol.}$ | 0.5 |
| Cauchy number | $C_Y$ | $2 \times 10^{-5}$ |
| Volume produced by foliages | $V_{prod.}$ | $4V_0 l$ |
| Maintenance thickness | $e$ | $0.02L$ |
| Forest radius | $R$ | $200L$ |
| Mutation probability | $p_{mut.}$ | 0.05 |
| Mutation amplitude | $\delta g$ | 0.005 |

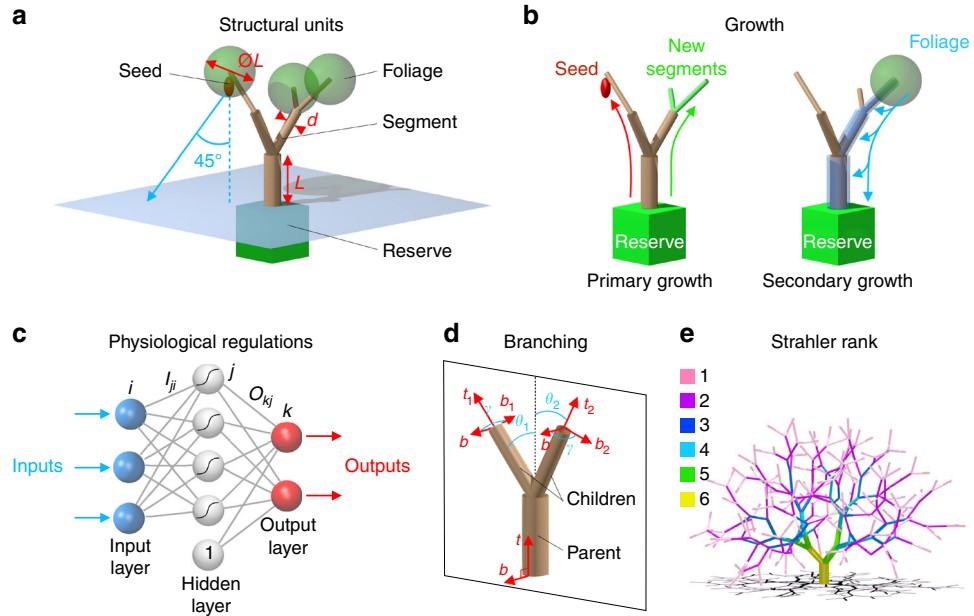

**Fig. 1** Principles of MECHATREE. **a** Each virtual tree is an assembly of different units: segments, foliages, seeds and a reserve. **b** Primary growth relies on the reserve to grow new segments and seeds. Secondary growth costs include maintenance in addition to diameter growth of each branch segment. **c** Formal neural networks are used to model the biochemical regulation of growth. **d** The growth of new segments follows a rule based on three angles: $\theta_1$, $\theta_2$ and $\gamma$. **e** Illustration of Strahler ordering of branches (Methods section)

(Table 1). These segments are connected at their extremities, such that each segment has a parent segment (except for the trunk), and 0, 1 or 2 child segments. Segments with no child, called twigs, are terminated by an assembly of leaves, called a foliage, modelled as a sphere of diameter $L$ centred on the segment distal end (Fig. 1a). Seeds can be produced at the twig ends. The reserve, whose exact location is not specified, stores assimilates from the current year that will be mobilised the following year to support primary and reproductive growth. Reserves have been included because of their significant impact on tree capacity to recover from major disturbances like strong wind damages.

**Growth.** The molecular regulations controlling growth strategies are implemented using formal neural networks. Here, neural networks are used as a tool that allows for an agnostic and flexible modelling of complex physiological regulations that do not involve actual neurons. Here, we make use of an important result known as the 'universal approximation theorem', which states that any continuous function can be approached with any prescribed accuracy provided the number of hidden neurons is large enough[40]. As illustrated in Fig. 1c, the artificial neural networks of MECHATREE consist of three layers: an input layer, a hidden layer and an output layer (for details, Methods section). Through the neuronal coefficients, these neural networks relate functionally the inputs to the outputs.

Growth processes are divided into 'primary growth' (the onset of new segments and seeds), and 'secondary growth' (the growth in diameter of existing segments, Fig. 1b). The strategy of primary growth is implemented with a 2–input, 3–hidden-neuron and 3–output neural network. The inputs are the volume of carbon contained in the reserve and the number of foliages in the tree. The outputs are a photosensitivity parameter and the proportions of carbon allocated to new segments and seeds. When new segments (children) of diameter $d_0 = 0.1L$ are added at the distal end of an existing segment (parent), geometrical rules inspired from the seminal models of Honda[41], and Niklas and Kerchner[42] are used (Fig. 1d, Methods section).

Secondary growth is implemented with a 2–input, 3–hidden-neuron and 1–output neural network. The inputs are the relative wind-induced stress felt by the segment, $\sigma_{max}/\sigma_0$ (Methods section), and the number of foliages irrigating the segment. The output is a safety factor $S$ accounting for the thigmomorphogenetic response (Methods section). The sink strength of a segment is the sum of the volume needed to achieve a certain safety against wind loads and a maintenance volume calculated as $V_{maint.} = \pi L d e$, with $e = 0.02L$ the thickness of the outer layer to be renewed every year. For every segment, this sink strength is equally partitioned among the foliages situated above in the hierarchy: each segment will 'request' a equal amount of photosynthates to the foliages above.

For each foliage, the photosynthates produced have a volume $V_{prod.} = 4V_0 l$, where $V_0 = \pi L d_0^2/4$ is the volume of newly grown twigs, and $0 \leq l \leq 1$ is the intercepted light calculated with a ray-tracing method[43] (Methods section). If the photosynthate volume produced by a foliage exceeds the total sink strength of the segments situated below, each segment receives its share and the leftover is stored in the reserve. On the contrary, if the volume produced is not sufficient, each segment receives photosynthates in proportion to its sink strength (Fig. 1b).

Tree growth is affected by both exogenous factors (wind, shade), and endogenous factors (branching angles, neural network coefficients). These endogenous factors are the 'genes' of a tree species, and together constitute its 'genome'. In the present case, there are 31 genes: 3 genes for the branching angles, and 18 and 10 genes for the coefficients of the primary-growth and secondary-growth neural networks respectively. These 31 genes are complemented with 3 'neutral marker genes' used for visualisation purposes.

Within this model, branch fall is provoked by mechanical loading, and this can occur along two different scenarios: either an extreme wind event occurs and branches can fracture with a probability described by a Weibull distribution (Methods section); or the foliage sources cannot provide enough photosynthates to ensure the maintenance costs of a given branch and

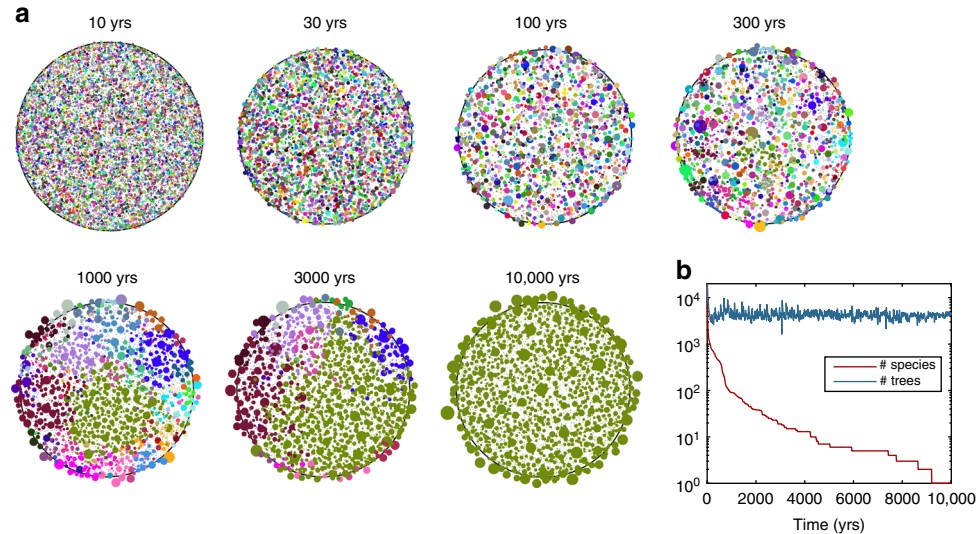

**Fig. 2** Evolution of a forest on a virtual island. **a** Example of the evolution of a virtual forest over 10,000 yrs (Supplementary Movie 1). Initially, 20,000 trees with random genomes are sown. Each circle is a tree whose centre is at the centre of gravity of foliages, and whose radius is the standard deviation of the foliage distribution projected onto the ground. The outer black circle of radius $R = 200L$ is the habitat limit. Colours are determined using three neutral marker genes as RGB values. Because trees growing at the periphery are on average larger than the others, allometric statistics (see below) are performed on trees with their trunk in a central zone of radius $0.9R$. **b** Number of trees and number of species as a function of time, for the same forest as in **a**

this branch will weaken over time to the point where it will fall down whatever the level of mechanical load.

**Competition**. The algorithm used to simulate the growth and evolution of a forest is divided into the following steps (steps 1–7 constitute a yearly cycle).

Step 0: Initialisation. In a circle of radius $R = 200L$, either 20,000 individuals with random genomes or 4000 individuals with selected genomes are sown at random locations. At this seeding stage, they are all formed of a single vertical segment of diameter $d_0 = 0.1L$ and a reserve of volume $2V_0$.

Step 1: Light interception. The sunlight intercepted by each foliage is calculated (Methods section).

Step 2: Stress calculation. The maximum bending stress, $\sigma_{max}$, is calculated in each segment (Methods section).

Step 3: Secondary growth. The photosynthates produced by foliages are allocated to maintenance and diameter growth in each segment.

Step 4: Pruning. A wind velocity is picked with random orientation and speed $U$ following an exponential distribution, such that the return period of wind speeds exceeding by 50% the average yearly maximum $U_0$ is 100 yrs[38]. The probability of pruning is then given by a Weibull distribution (Methods section).

Step 5: Death. Trees die when one of these two conditions is realised: (i) their age is larger than 1000 yrs; (ii) their age is larger than 6 yrs and their number of segments is less than 10.

Step 6: Primary growth. The carbon stored in the reserve is allocated to grow seeds and new segments.

Step 7: Reproduction. Seeds fall with a 45° angle with the vertical and form single-segment trees with the same genome as their parent except for slight mutations (Methods section).

With the goal of identifying the best-adapted growth strategies in a competitive environment, a single-elimination tournament is run. During the first round, the growth and evolution of 32 different forests is simulated. Each forest is initialised with 20,000 trees with random genomes. The natural selection of a tree phenotype (and genotype) within a forest is a 'game' that yields a single or few 'winners', i.e. species that dominates all others

(Fig. 2). Although trees of the same species have a common ancestor, their genomes differ slightly, because of mutations at each generation. After these simulations have been run for 10,000 yrs, the genomes of the 2000 oldest trees, the 'winners', are collected in each of the 32 forests.

In the 2nd round, the growth of 16 forests is simulated. These forests are now initialised with 4000 trees, composed of the winners of two first-round games. After 20,000 yrs, the 2000 winners are again collected in each game. This operation is repeated at each round until the Final is reached (Supplementary Fig. 1). During the Final, the overall winning species are identified. Among the initial 0.64 million random genomes, these overall winners are the 'fittest': not only have they survived more than 200,000 yrs in a competing environment, but the slight mutations at each generation have allowed their genomes to adapt. This highly simplified evolutionary process makes it possible to reach a meaningful growth strategy without a priori knowledge.

In the Final, after a transient, two different species eventually coexist. We have performed simulations over more than 1 million year, and there is no sign of one of these species becoming extinct. These two species are associated to two different ecological niches: the periphery and the interior of the island (Supplementary Fig. 1). In the Supplementary Discussion, we analyse how the allocation strategies of these two species differ (see also Supplementary Figs. 2 and 3). However, when simulations are ran with smaller islands ($R = 40L$ or $100L$ instead of $R = 200L$), after a few thousand years, only the periphery species remains. For convenience, in the following we will refer to this periphery species as the 'fittest species'.

We will now determine if the trees selected by present model exhibits allometric scalings similar to those observed empirically. First, competition for light can be assessed through self-thinning, which is the relation between stand density and average tree biomass. Second, interspecific tree allometry can be examined for the species that have survived the first 3000 yrs of simulation. Finally, for the fittest species, self-similarity is assessed by examining different allometric scalings within an architecture: self-similar ratios, tapering law and area conservation.

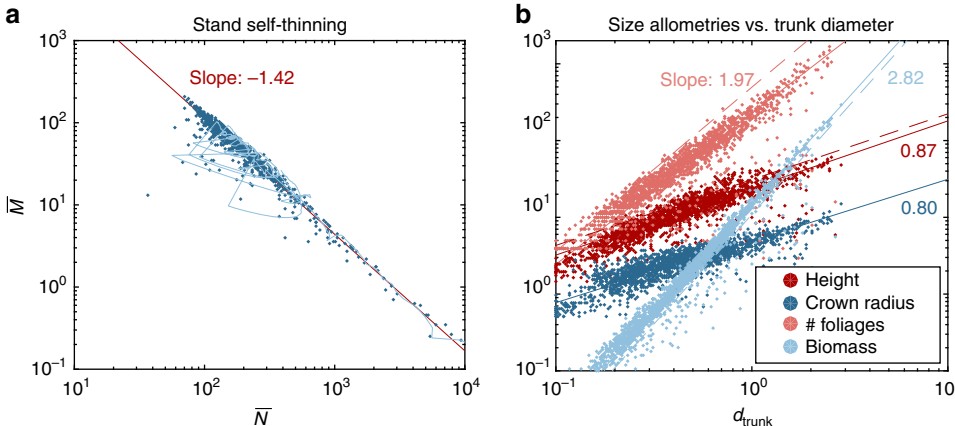

**Fig. 3** Allometric scalings. **a** For the 32 forests of the first round, the effective biomass $\overline{M}$ (unit $L^3$) is plotted as a function of the effective number of trees $\overline{N}$, for the first 500 yrs (the light blue line shows the history of a particular forest). Only 2% of the dataset is shown, but the red line shows a linear regression fit on the entire dataset with $\overline{N}$ as weight. Large excursions to the left of the regression line correspond to strong wind events during which a large number of trees can die (Fig. 2b and Supplementary Movie 1). **b** Each tree in the 32 forests of the first round is extracted after 3000 yrs. Their height $H$ (unit $L$), crown radius $C$ (unit $L$), number of foliages $N$ and stem biomass $B$ (unit $L^3$) are plotted as a function of their trunk diameter $d_{trunk}$. For clarity, only 5% of the trees are shown, but the solid lines show reduced major axis regression (RMA) on the whole dataset, with $N$ as weight. The dashed lines show the results of the AMT model (see below)

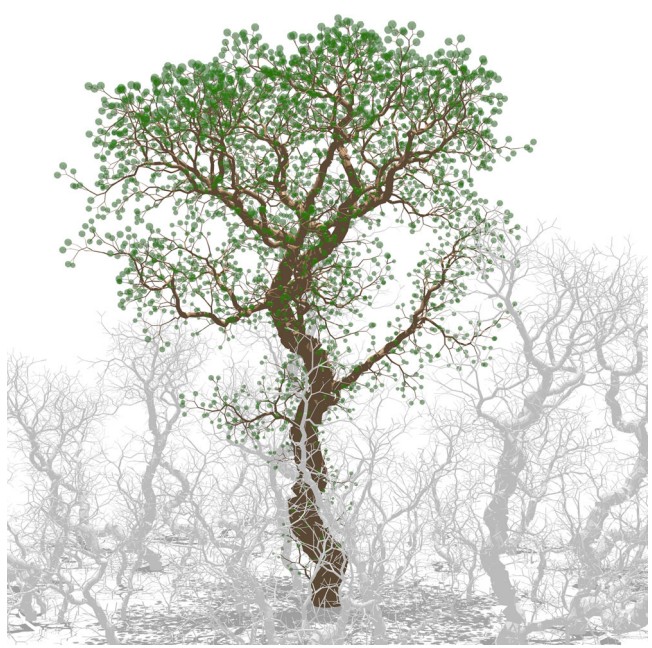

**Fig. 4** Partial view of the forest in the Final round on a small island ($R = 40L$), when only the fittest species remains. Only the largest tree has been coloured for clarity. In this representation, the diameter of foliage spheres is proportional to the light intercepted (Supplementary Movie 2)

**Self-thinning**. The empirical $-3/2$ self-thinning law for plant populations states that the average biomass of plants decreases as $n^{-3/2}$, with $n$ the plant density[44]. This relation can be assessed by computing how the 'effective number' $\overline{N}$ and 'effective biomass' $\overline{M}$ vary with time[45] (Methods section). In Fig. 3a, these quantities are plotted for the initial 500 yrs of the first-round forests. Initially $\overline{N} = 2 \times 10^4$ and $\overline{M} = V_0 \approx 0.008\,L^3$, then, as trees grow, $\overline{M}$ increases and $\overline{N}$ decreases following an allometric relation: $\overline{M} \propto \overline{N}^{\beta_{ST}}$, with $\beta_{ST} \approx -1.418$ (95% confidence interval, CI: $-1.421$–$1.416$). The present model is thus in agreement with the empirical self-thinning law (i.e. $\beta_{ST} \approx -3/2$), showing that, in MECHATREE, competition for light, growth and mortality are consistent with empirical observations for young forests.

**Tree allometry**. In Fig. 3b, trees found in first-round forests are compared. Height $H$, crown radius $C$ (measured as the standard deviation of the foliage distribution projected onto the ground, as in Fig. 2a), number of foliages $N$ and stem biomass $B$ are plotted as a function of the trunk diameter $d_{trunk}$ for each tree. It shows that the ~1000 different species that have survived the initial 3000 yrs exhibit allometric relations: $H \propto d_{trunk}^{\beta_H}$ with $\beta_H \approx 0.87$ (95% CI: 0.871–0.876), $C \propto d_{trunk}^{\beta_C}$ with $\beta_C \approx 0.80$ (95% CI: 0.802–0.805), $N \propto d_{trunk}^{\beta_N}$ with $\beta_N \approx 1.97$ (95% CI: 1.966–1.971), $B \propto d_{trunk}^{\beta_B}$ with $\beta_B \approx 2.82$ (95% CI: 2.822–2.825). Similar exponents are found when the forests are composed of the two finalist species (Supplementary Fig. 4) and a sensitivity analysis shows that these exponents depend only weakly on the model parameters (see below).

From these exponents, different experimental allometric relations can be recovered. First, it can be seen that the biomass roughly follows the classical scaling $B \propto H d_{trunk}^2$. Then, assuming that the total mass of leaves, $M_L$, scales as the number of foliages $N$, ones finds that $M_L \propto B^{\beta_{ML}}$ and $M_L \propto d_{trunk}^{\beta_N}$, with $\beta_{ML} = 0.68$ and $\beta_N = 1.97$. These values are similar to the exponents measured on angiosperms and gymnosperms: $\beta_{ML} = 0.75$ and $\beta_N = 2.17$[5] (they are about 9% smaller). In addition, the distribution of trunk diameters in the forests composed of the finalist species scales as $d_{trunk}^{-2}$ (Supplementary Fig. 5). This scaling is the same as the one generally observed in forest communities, which has also been predicted by the theory of metabolic ecology[46, 47].

These allometric scalings emerge in MECHATREE because the trees selected through the single-elimination tournament share common characteristics. All have a similar safety factor against wind loads $S \approx 3$ (Supplementary Fig. 6c), and their architecture is self-similar with a fractal dimension around $D \approx 2.5$ (Supplementary Fig. 6e). Finally, in the literature, different values of the allometric exponent have been reported in the interval $0.54 \leq \beta_H \leq 0.89$[48–50]. The exponent $\beta_H = 2/3$ has also been predicted based on arguments of elastic similarity[4]. However, some authors argued that the relation between the logarithms of tree height and trunk diameter is not linear but curvilinear because of finite-size effects[11, 51, 52]. The same trend is observed in our data: $\beta_H$ tends to decrease with $d_{trunk}$ (Fig. 3b). Another reason for curvilinearity is that young trees are not self-similar (see below).

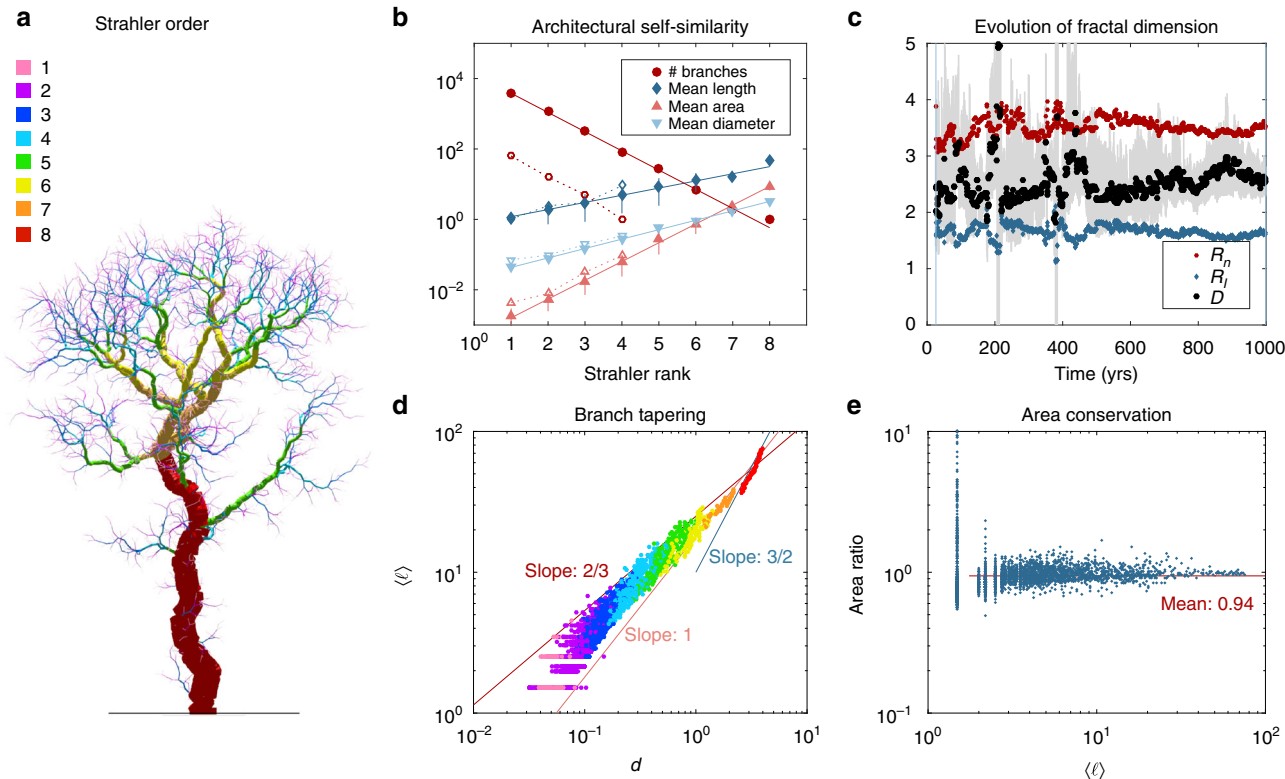

**Fig. 5** Self-similarity. **a** Representation of the Strahler order of each branch for the tree illustrated in Fig. 4. **b** The number of branches, their mean length (unit $L$), area (unit $L^2$) and diameter (unit $L$) are plotted as a function of their Strahler order for the 999-year-old tree illustrated in Fig. 4 and in **a**. Error bars show standard deviations around these means. Solid lines are regression fits on the first 7 ranks. Open symbols connected by dotted lines represent the same quantities when the tree is only 25 years old. **c** Evolution of the branching ratio $R_n$, length ratio $R_l$ and fractal dimension $D = \ln R_n / \ln R_l$ during the lifetime of the same tree. Grey bars show the 80% confidence interval for $D$. **d** Branch tapering is illustrated by plotting, for each segment, the average distance from the foliages, $\langle \ell \rangle$, as a function of the diameter $d$. The Strahler order of each segment is represented with the same colour code as in **a**. **e** Assessment of Leonardo's rule of area conservation across branching nodes. For each node, the area ratio (i.e. the ratio between the total cross-sectional area of children segments and the parent area) is plotted as a function of the average distance $\langle \ell \rangle$ of the parent segment from the foliages. The average area ratio measured for $\langle \ell \rangle > 1.5$ is 0.94. When restricted to $\langle \ell \rangle > 10$, the average is 0.985

**Self-similar ratios**. In Fig. 4, a forest on a small island ($R = 40L$) is depicted when only the fittest species remains. Colouring the branches of the tallest tree according to their Strahler ranks shows qualitatively that the larger the rank, the thicker and the longer the branches (Fig. 5a). A more quantitative analysis is performed by examining the number of branches $n_k$, their mean length $l_k$, diameter $d_k$ and cross-sectional area $a_k$, as a function of their rank $k$ (Fig. 5b). These quantities follow a geometric progression such that four self-similar ratios can be defined: the branching ratio $R_n = n_k/n_{k+1}$, the length ratio $R_l = l_{k+1}/l_k$, the diameter ratio $R_d = d_{k+1}/d_k$ and the area ratio $R_a = a_{k+1}/a_k$. A linear regression shows that $R_n = 3.51$ (95% CI: 3.42–3.60), $R_l = 1.60$ (95% CI: 1.52–1.67), $R_d = 1.85$ (95% CI: 1.81–1.90) and $R_a = 3.41$ (95% CI: 3.23–3.61).

The existence of branching and length ratios, $R_n$ and $R_l$, both independent of $k$, means that the tree skeleton is self-similar and that a fractal dimension can be defined: $D = \ln R_n / \ln R_l = 2.68$ (95% CI: 2.39–3.04). This fractal dimension is a measure of how the tree structure fills space. As can be seen in Fig. 5c, fractal dimension varies with time. When the tree is very young, $D$ is not well defined, mainly because branch lengths do not progress geometrically with Strahler order (see open symbols in Fig. 5b). Then, as the tree grows and ages, $D$ fluctuates in the interval $2 < D < 3$. Note that the amplitude of these fluctuations is much lower for a tree growing without competitors (Supplementary Figs. 7–10). It can be noted that $R_l \approx R_d$, which means that branch aspect ratio does not vary to a great extent ($9 < l_k/d_k < 25$).

**Table 2 Self-similar ratios**

|  | $R_n$ | $R_l$ | $R_d$ | $D$ |
|---|---|---|---|---|
| Virtual tree in forest (Fig. 4) | 3.51 | 1.60 | 1.85 | 2.68 |
| Virtual tree alone (Supplementary Fig. 7) | 3.50 | 1.74 | 1.88 | 2.26 |
| Red Oak[4] | 3.83 | – | 1.56 | – |
| Poplar[4] | 4.22 | – | 1.86 | – |
| Fir[3] | 4.8 | 2.7 | – | 1.6 |
| Apple Tree[68] | 4.35 | – | 1.90 | – |
| Birch tree[68] | 4.00 | – | 1.94 | – |
| Pinyon pine[69] | 3.63 | 1.71 | 1.81 | 2.40 |

The ratios $R_n$, $R_l$, $R_d$ and fractal dimension $D$ for the virtual trees of Fig. 4 and Supplementary Fig. 7 are compared with empirical data of the literature

Another property is that $R_n \approx R_a$, meaning that the total cross-section of branches is independent of the rank. This rule of area conservation can also be established with a different method (see below).

Self-similar ratios and fractal dimensions have been rarely measured in individual trees (Table 2). So far, the results of the present model ($3 < R_n < 4$, $1.5 < R_l < 2$ and $2 < D < 3$, Fig. 5c) seem consistent with the available data. Not only the growth strategy developed by the fittest virtual species yields a self-similar architecture, but its quantitative characteristics resemble empirical observations.

**Tapering**. Tapering describes, through an allometric relation, how the diameter $d$ of a segment is related to its distance $\ell$ from the branch apex: $d \propto \ell^{\beta 4}$. In a ramified structure though, $\ell$ is not unique and is replaced by $\langle \ell \rangle$, the average distance of all paths connecting a segment to its descendant foliages. In Fig. 5d, $\langle \ell \rangle$ is plotted as a function of $d$, for the tree of Fig. 4. This plot is very similar to what has been measured on trees (e.g. Fig. 6 in ref. [4].). It exhibits different regions: for small $d$, the scatter is important and data do not follow a simple allometric relation; for intermediate $d$, the allometric relation $d \propto \langle \ell \rangle^{\beta}$ with $\beta \approx 3/2$ seems to be reached; for large values of $d$ corresponding to the trunk, $\beta \approx 2/3$, which is the exponent expected for uniformly distributed wind-induced loads along the trunk height[14]. This is likely due to the fact that the 'sail area' of the trunk is of the same order as the sail area of all foliages.

**Area conservation**. In his notebooks, Leonardo da Vinci observed that 'all the branches of a tree at every stage of its height when put together are equal in thickness to the trunk'[9]. In other words, according to Leonardo's rule, the cross-sectional area should be conserved across branching nodes on average. One way to assess the validity of this rule is to plot for every branching node, the area ratio $(a_1 + a_2)/a_0$, where $a_1$ and $a_2$ are the cross-sectional areas of children branches and $a_0$ is the area of the parent branch. A recent study has shown that the average value of this ratio is between 0.90 and 1.05 for five species: Balsa, Maple, Oak, Pinyon and Ponderosa pine[53]. In MECHATREE, the same assessment of Leonardo's rule has been performed and is shown in Fig. 5e. This plot is very similar to the measurements made on real trees (e.g. Fig. 3a in ref. [53].). In both cases, the average value of the area ratio is close to 1, which is consistent with Leonardo's rule of area conservation.

**Sensitivity analysis**. By design, MECHATREE involves as few parameters as possible for a process-based model (Table 1). Some of these parameters, related to wind-induced loads, maintenance costs and photosynthesis, have been chosen to match empirical observations. In particular, the Cauchy number has been fixed to $C_Y = \rho U_0^2 / \sigma_0 = 2 \times 10^{-5}$, with an air density $\rho = 1.2$ kg m$^{-3}$, an average yearly maximal wind $U_0 = 40$ m s$^{-1}$ and wood strength $\sigma_0 = 100$ MPa[54]. The maintenance thickness is taken equal to $e = 0.02 L$, or, by taking $L = 10$ cm, is equal to $e = 2$ mm, the typical thickness of the inner bark. Taking a foliage with a leaf surface of 150 cm$^2$ [55], which is about half the area of a sphere of diameter $L = 10$ cm, the optical transparency is set to $\alpha_{\text{fol.}} \approx 0.5$. With a dry wood density of $\rho_{\text{wood}} = 613$ kg m$^{-3}$ [56, 57], $V_0$ corresponds to a dry mass of 4.8 g. Taking a typical net assimilation rate of leaves of 10 g m$^{-2}$ per day[58, 59], each foliage produces during 6 months about 27.3 g of photosynthates. This corresponds to $V_{\text{prod.}} \approx 4V_0 l$ (assuming 70% of full light on average, i.e. $l = 0.7$).

To test the robustness of the allometric laws, we have conducted a sensitivity analysis on four model parameters: $C_Y$, $e$, $\alpha_{\text{fol.}}$ and $V_{\text{prod.}}$. Model parameters have been varied, one at a time, in a $\pm 30\%$ interval around their reference values given in Table 1. Each time, 16 simulations are performed in forests of radius $R = 200L$. After 3000 yrs, the data of all trees are collected and RMA regressions are performed to identify the allometric exponents and intercepts (Fig. 3b), together with the typical safety factor $S$, and an average fractal dimension $D$ (see section 'Sensitivity analysis on the allometric laws' in Methods section).

The main result is that, although trees grow to a larger size when $C_Y$ or $e$ are decreased, or when $V_{\text{prod.}}$ or $\alpha_{\text{fol.}}$ are increased, the allometric exponents $\beta_N$, $\beta_B$ and $\beta_{\text{ML}}$ do not vary significantly. The only exception is the dependence of $\beta_H$, which is likely due to a curvilinear relation between the logarithms of $H$ and $d_{\text{trunk}}$

(Fig. 3b). The allometric laws emerging from MECHATREE are therefore robust to variations of the model parameters.

An interesting result of the sensitivity analysis is the dependence of the fractal dimension on the foliage transparency. This result is also confirmed by a parametric analysis that consists in comparing how an isolated tree of the fittest species grow when the model parameters are varied (Supplementary Methods). Both analyses show that fractal dimension increases with foliage transparency and this can be interpreted as follows. The outer surface covered by foliage clusters generally has a dimension around 2 (it can be slightly larger if it has some fractal roughness). When foliage clusters are opaque, foliage inside this surface does not intercept any light and will eventually be shed because the branches supporting them do not have the resources to ensure maintenance costs. Since foliages and the structure supporting them have generally the same dimension, we expect $D \approx 2$ for opaque foliages. This contrasts with the case of fully transparent foliages, where the structure is expected to be volume-filling (i.e. $D = 3$). Owing to the central role of chlorophyll in both photosynthesis and leaf transmittance properties, $V_{\text{prod.}}$ and $\alpha_{\text{fol.}}$ are likely to be negatively correlated. Whether there is an optimal transparency remains, however, an open question.

**The AMT model**. Genotypes that survive the initial 3000 yrs of simulation exhibit allometric relations close to the ones observed in nature (Table 3). Interestingly, these relations can be recovered with a simple analytical model that we shall call the AMT (Analytical MechaTree) model. The AMT model is thus a way to capture the minimal set of factors explaining the emergence of allometric scalings in MECHATREE. It is also useful to compare our results with other models based on an analytical approach, such as the WBE model[21].

To build the AMT model, we use the three emergent results of MECHATREE: (i) tree architectures (skeletons) are self-similar; (ii) their fractal dimension is $D \approx 2.5$; (iii) their safety against wind loads is constant. If the tree skeleton is self-similar, branching and length ratios are independent of the rank $k$, and a fractal dimension $D$ can be defined such that $R_n = R_l^D$. Consider now a branch at rank $k$. It is fed by the phloem sap flows coming from $N_k = R_n^{k-1}$ foliages, which are at an average distance $\langle l \rangle_k = L(R_l^k - 1)/(R_l - 1)$ from the branch base. As a result, the relative wind-induced bending stress is $\sigma_k/\sigma_0 = \frac{16\alpha}{\pi} C_Y S_{\text{fol.}} N_k \langle l \rangle_k / d_k^3$, where $\alpha$ is an order 1 geometrical parameter that accounts for the angle between wind and branches. If the safety factor $S$ is constant (Assumption iii), branch diameters will be such that $\sigma_k/\sigma_0 = S^{-3/2}$. With these arguments, the diameter ratio is found to be $R_d = R_n^{(D+1)/3D}$, and the allometric exponents are (see Methods section, for details).

$$\beta_H = \frac{3}{D+1}, \quad \beta_N = \frac{3D}{D+1}, \quad (1a)$$

$$\beta_B = \frac{2D+5}{D+1}, \quad \beta_{\text{ML}} = \frac{3D}{2D+5} \quad (1b)$$

Remarkably, within this simple analytical model based on geometrical and mechanical arguments, allometric exponents only depend on $D$. Using the value $D = 2.5$ in the previous analytic formulae (Assumption ii) yields allometric exponents in good agreement with the results of the numerical simulations of virtual forests (Table 3). This is consistent with the fractal dimension $2 < D < 3$ that has been computed independently.

Comparison with measurements on angiosperms and gymnosperms[5, 6, 50] shows also an excellent agreement (Table 3,

**Table 3 Comparison of allometric exponents**

|  | $\beta_H$ | $\beta_N$ | $\beta_B$ | $\beta_{ML}$ |
|---|---|---|---|---|
| Virtual forests | 0.87 | 1.97 | 2.82 | 0.68 |
| AMT model | 0.86 | 2.14 | 2.86 | 0.75 |
| WBE model | 0.67 | 2 | 3 | 0.75 |
| SERA model | 0.86 | 1.98 | 2.66 | 0.74 |
| Dicots, conifers | 0.73 | 2.17 | 2.89 | 0.74 |
| (95% CI) | (0.71–0.76) | (2.01–2.32) | (2.71–3.14) | (0.738–0.742) |

Tree height $H$, leaf mass $M_L$, stem biomass $B$ and trunk diameter $d_{trunk}$ are related through allometric relations: $H \propto d_{trunk}^{\beta_H}$, $M_L \propto d_{trunk}^{\beta_N}$, $B \propto d_{trunk}^{\beta_B}$ and $M_L \propto B^{\beta_{ML}}$. The allometric exponents found for the virtual forests, the AMT model ($D = 2.5$), WBE model ($D = 3$)[21], the SERA model for angiosperms[36] are compared to empirical observations on dicots and conifers[5, 6, 50], given with their 95% confidence intervals

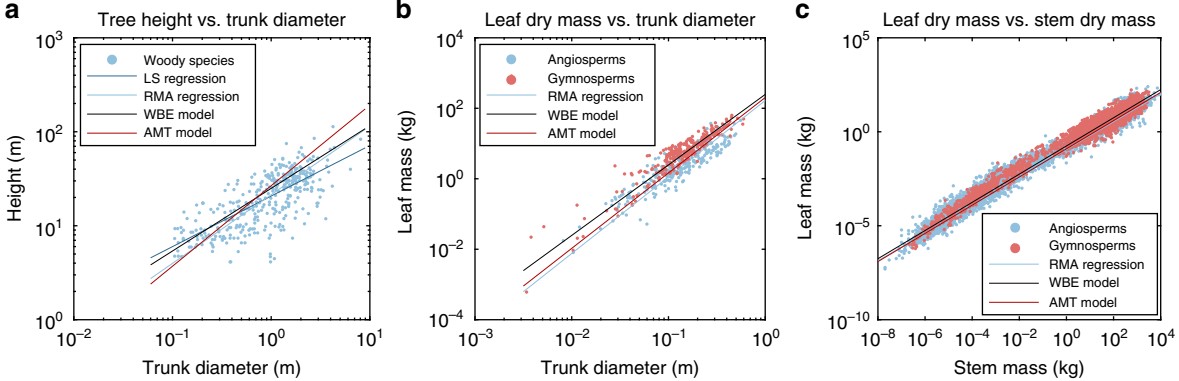

**Fig. 6** Comparison of the AMT model with empirical data. **a** Scaling of plant height as a function of stem diameter. The allometric data of ref. [48] for woody species are compared to least square (LS) regression $H = 20.7\,d_{trunk}^{0.538}$[48], reduced major axis (RMA) regression $H = 21.4\,d_{trunk}^{0.73}$[50], WBE model $H = 25\,d_{trunk}^{0.67}$ and present AMT model $H = 26.7\,d_{trunk}^{0.857}$ (Methods section and Supplementary Discussion). **b** Allometric data on leaf dry mass vs. trunk diameter[5] are compared to RMA regression $M_L = 166\,d_{trunk}^{2.17}$[5], WBE model $M_L = 247\,d_{trunk}^2$ and AMT model $M_L = 202\,d_{trunk}$. **c** Allometric relations between leaf dry mass and stem dry mass from more than 11,000 records[6] are compared to RMA regression $M_L = 0.113\,M_S^{0.74}$[6], WBE model $M_L = 0.176\,M_S^{0.75}$ and AMT model $M_L = 0.124\,M_S^{0.75}$

Fig. 6). Besides, the value of the fractal dimension ($D \approx 2.5$) is compatible with measurements on tree structures (Table 2) and tree crowns ($2.13 < D_{fol.} < 2.76$[60]), which should be similar to the dimension of the branch architecture. Note that the AMT model correctly predicts not only the allometric exponents, but also the allometric intercepts (Fig. 6 and Supplementary Discussion). Note also that agreement is excellent over a large range of trunk diameters or stem dry mass (Fig. 6). This is rather surprising because the model should only provide realistic predictions for trees that are large enough to have a properly defined fractal dimension and small enough such that gravity loads remains negligible.

## Discussion

In this paper, a generic functional–structural model of tree, called MECHATREE, has been developed. This model has been used to address the best growth strategy of trees when they compete for light and are subject to wind-induced loads. Thanks to its relative simplicity, MECHATREE allowed us to simulate entire forests over long periods of time. This feature has been exploited by running a single-elimination tournament to identify the fittest virtual species in this simplified environment. The self-similar and allometric properties of this species have then been quantified and found to be similar to those of existing tree species.

The existence of branching, length and diameter ratios shows that tree architectures are self-similar (more rigorously, they are self-affine since the different lengths can vary differently with scale). Yet, the whole structure is an assembly of segments of exact same length. Self-similarity is thus an emergent property resulting from the complex architecture reorganisations that occur during the lifetime of a tree through wind-induced pruning and light- and wind-dependent growth. Two important results can be deduced from MECHATREE. First, self-similarity of the tree skeleton and the existence of a fractal dimension $D$ mainly emerge from competition for light. As a result, $D$ is strongly linked to foliage transparency. Second, for realistic probabilities of extreme wind events, the safety factor against wind loads is approximately constant. This efficient strategy to resist winds leads to self-similarity of the branch diameters and Leonardo's rule of area conservation.

Classical allometric laws relating a tree's trunk diameter to its height, stem biomass or leaf biomass have been recovered with the present numerical model and with a simplified analytical model we developed, the AMT model. Alternative models, such as the SERA model[36] or WBE-related models[5, 21, 22] can also predict allometric exponents close to the ones measured on trees (Table 3). Comparison with the present models is interesting, but meaningful only when the nature of the models is similar.

The SERA model is an individual-based model with species-specific parameters describing how an individual tree grows when it competes for space and light[36]. From this model, allometric relations emerge at the population level (Table 3). In that sense, it bears some similarity with the present study. There are two major differences between MECHATREE and SERA however: MECHATREE does not involve species-specific parameters other than those selected by the simulated evolution; and the modelling units are the branch segments, which allows us to address the

origin of self-similarity in individual trees, which is not possible in SERA.

The WBE model is an analytical model based on geometrical, mechanical and hydraulic arguments. In particular, it assumes that trees are volume-filling fractal networks (i.e. $D = 3$). It can be used to derive allometric relations that are very similar to the predictions of the AMT model (Fig. 6, Supplementary Table 1 and Supplementary Discussion). It is thus impossible to refute either approach based on allometry only. One way to achieve falsification would be to have more data on the fractal dimension, which should be possible with the development of Lidar-based technologies[61], or to directly assess the mechanistic bases of the processes involved, e.g. through wind mechanosensing[62].

In the WBE model[21], or related models[22], mechanics is implemented by enforcing an elastically similar allometric relation between axis lengths and radii[4] (Supplementary Discussion). In MECHATREE, the mechanical stresses and the species-specific thigmomorphogenetic response are calculated for each segment every year. Self-similarity and allometry are not imposed, but emerge from thigmomorphogenesis and competition for light, two major processes in temperate deciduous forests[62, 63].

The importance of the various hydraulic traits (water use efficiency, embolism) and mechanical traits (wind hazards, self-weight) may vary between species habitats and plant stages. In some cases, hydraulic performance may be a major selective pressure, whereas in others wind mechanical safety will be dominant. Yet these different situations may not be identified in broad range allometric data, since both selective pressures yield similar allometries. We may however speculate that the presence of both selective pressures can increase the speed of natural selection. We believe that further insights could be gathered by combining the biomechanically-based AMT model proposed in the present paper with the hydraulic hypotheses of the initial WBE model[23] to have a better understanding of both mechanics and hydraulics.

## Methods

**Formal neural networks.** The artificial neural networks used to model agnostically the growth strategies of different species consist of an input layer of neurons where the stimuli arrive with intensity $x_i$ (normalised to be of order 1). These stimuli are linearly combined and sent to a hidden layer of neurons receiving a signal $y_j$, with $y_j = \sum_i I_{ji} x_i$. The hidden-layer neurons then perform a nonlinear transformation of the signal: $y'_j = \tanh(5y_j)$ (except for one neuron not linked to the input layer and sending a unit signal). Finally these signals are linearly combined and transmitted to the output layer with intensity $z_k$, with $z_k = \sum_j O_{kj} y'_j$ (Fig. 1c).

**Primary growth.** Carbon is converted into primary growth through constructions costs: a new segments costs its volume $V_0$ (Table 1) and a seed costs $5V_0$, which accounts for the volume of its initial reserve, $2V_0$, the volume of the first sprout, $V_0$, and a supplementary cost for dissemination and germination ($2V_0$).

From a practical viewpoint, each year, the primary growth neural network computes for each tree three outputs: the proportions of carbon allocated to new segments ($0 \leq P_{\text{seg.}} \leq 1$) and to seeds ($0 \leq P_{\text{seed}} \leq 1$), with $P_{\text{seg.}} + P_{\text{seed}} \leq 1$, and a photosensitivity parameter, $0 \leq p \leq 1$. If $V_{\text{res}}$ is the volume of the reserve, a portion of it is allocated to the construction of $n_{\text{seg.}} = \text{floor}(P_{\text{seg.}} V_{\text{res}}/V_0)$ new segments and another portion to the construction of $n_{\text{seed}} = \text{floor}(P_{\text{seed}} V_{\text{res}}/5V_0)$ seeds. Both new segments and seeds are located according to the photosensitivity parameter $p$: for $p = 0$, locations are picked at random among twigs; for $p = 1$ the most lit foliages are selected.

**Light interception.** To calculate the light $l$ intercepted by each foliage, a ray-tracing numerical method is used[43]. It is assumed that shadow cast by segments can be neglected, and that sunlight is uniformly distributed in the upper hemisphere, which is divided into 32 equal solid angles. For each solid angle, the whole forest is rotated, such that $z$ is along the mean direction of the solid angle. Foliages are then ordered by descending $z$ and the $(x, y)$ position of each foliage is discretised onto a grid of $L \times L$ squares. When different foliages belong to the same grid square, the highest one receives 1/32 of light, the second highest $\alpha_{\text{fol.}}/32$, the third $\alpha_{\text{fol.}}^2/32$, etc., where $\alpha_{\text{fol.}} = 0.5$ is the optical transparency of foliages[43, 55]. This procedure is repeated for each of the 32 solid angles and gives a good approximation of the total light interception with a numerical scheme that scales as $N \log(N)$, with $N$ the

number of foliages. Note that the slowest part of this algorithm is to sort foliage heights, for which a quicksort algorithm is used[64].

**Branching angles.** New child segments are added by following geometrical rules based on three angles: $\theta_1$, $\theta_2$ and $\gamma$[9, 41, 42] (Fig. 1d). For a given parent segment oriented in the direction of the unit vector **t** and normal to the unit vector **b** (for the trunk, **b** is a random unit vector in the horizontal plane), two child segments of length $L$ are constructed in the plane normal to **b**. Their tangential unit vectors $\mathbf{t}_1$ and $\mathbf{t}_2$ are obtained by rotating **t** with the angles $\theta_1 + \epsilon\delta\theta$ and $\theta_2 + \epsilon\delta\theta$ around **b**. For a given tree, angles $\theta_1$ and $\theta_2$ are two constants, $\delta\theta = 10°$, and $\epsilon$ is a normal random variable with standard deviation 1. The normal vectors $\mathbf{b}_1$ and $\mathbf{b}_2$ defining the next-generation branching planes are obtained by rotating **b** with an angle $\gamma + \epsilon\delta\theta$ around $\mathbf{t}_1$ and $\mathbf{t}_2$, respectively.

**Genome and mutation.** During the lifetime of a tree, its genome does not vary. However, the seeds produced by a tree have a mutated genome. Mutation rules are as follows: with a probability $p_{\text{mut.}} = 0.05$ each gene $g$ is replaced by $g + \epsilon\delta g$, with $\delta g = 0.005$ the amplitude of the mutation and $\epsilon$ a random normal variable of unit standard deviation. Naturally, if $g$ happens to be below 0 or above 1 after the mutation, it is set to 0 or 1, respectively. The values of $p_{\text{mut.}}$ and $\delta g$ have been chosen to ensure that moderate mutations can occur over 10,000 yrs, the typical time scale of simulations.

**Wind-induced stress.** The wind-induced stresses in a tree are calculated assuming a uniform wind velocity $U\mathbf{u}$, where **u** is a horizontal unit vector. The wind-induced force on each foliage is then $\mathbf{F}_{\text{fol.}} = \frac{1}{2}\rho U^2 S_{\text{fol.}} \mathbf{u}$, where $\rho$ is the air density and $S_{\text{fol.}} = 0.25L^2$ is the 'sail area' of foliages in strong winds. Here a drag coefficient of 1 has been assumed without loss of generality. In addition, the wind exerts also a force on each segment. If **n** is the unit vector normal to both the wind and the segment (i.e. $\mathbf{n} = \mathbf{t} \times \mathbf{u}/\|\mathbf{t} \times \mathbf{u}\|$) the force exerted on each segment is $\mathbf{F}_{\text{segment}} = \frac{1}{2}\rho U^2 dL\|\mathbf{t} \times \mathbf{u}\|^2 \mathbf{n} \times \mathbf{t}$, where the drag coefficient is also taken to be 1, $d$ and $L$ are the diameter and length of the segment. This force is applied on the segment centre of mass and its moment at the segment base is simply $\mathbf{M}_{\text{segment}} = \frac{1}{2}L \mathbf{t} \times \mathbf{F}_{\text{segment}}$.

Now each segment transmits the forces and moments applied at its extremity to its base. If $\mathbf{F}_{\text{top}}$ and $\mathbf{M}_{\text{top}}$ are the sum of forces and moments at a segment top, force and moment at the base are

$$\mathbf{F}_{\text{base}} = \mathbf{F}_{\text{segment}} + \mathbf{F}_{\text{top}}, \tag{2a}$$

$$\mathbf{M}_{\text{base}} = \mathbf{M}_{\text{segment}} + \mathbf{M}_{\text{top}} + L \mathbf{t} \times \mathbf{F}_{\text{top}}. \tag{2b}$$

The moment at the base $\mathbf{M}_{\text{base}}$ has a bending component of intensity $M = \|\mathbf{M}_{\text{base}} \times \mathbf{t}\|$. The corresponding maximal bending (tensile and compressive) stress occurs at the surface and is $\sigma = \frac{32}{\pi} M/d^3$. To compute the maximal bending stress $\sigma_{\max}$ sensed by segments, wind speed is assumed to be constant, $U = U_0$, but different orientations separated by 45° angles are considered.

**Fracture probability.** The probability of fracture of a given segment can be modelled by a Weibull distribution to take into account volume effects[65]. This probability reads

$$P(\sigma, V) = 1 - \exp\left[-\frac{V}{V_0}\left(\frac{\sigma}{\sigma_0}\right)^m\right], \tag{3}$$

where $V$ is the volume of the segment, $\sigma$ is the bending stress in the segment, $\sigma_0$ is the strength of the material, $V_0$ is a reference volume taken to be the volume of new twigs, and $m = 10$ is the typical Weibull's modulus for wood[54]. Since the wind-induced bending stress is proportional to $\rho U_0^2$, with $U_0$ the average yearly-maximal wind, the probability of fracture is a function of a dimensionless Cauchy number

$$C_Y = \rho U_0^2/\sigma_0. \tag{4}$$

In particular, $P$ does not depend on the typical size $L$. Considering an average yearly-maximal wind of $U_0 \approx 40\,\text{ms}^{-1}$ and a wood strength $\sigma_0 \approx 100\,\text{MPa}$[54], the typical value of the Cauchy number is $C_Y = 2 \times 10^{-5}$.

The safety factor $S$ used to implement the thigmomorphogenesis response is such that the segments aims to reach a volume $V = SV_{\text{fract.}}$, with $V_{\text{fract.}}$ the volume giving $\sigma_{\max} = \sigma_0$.

**Strahler order.** A topological rank can be assigned to each segment, following a method originally developed by Strahler for river networks[66]. Within this framework, a 'branch' is defined as an assembly of contiguous segments of same rank. The principle of this ordering scheme is to assign the rank 1 to all terminal branches (i.e. assemblies of segments starting at the twigs and ending at the first branching node). Supposing that these rank 1 branches are then removed, the new terminal branches are assigned the rank 2, and so on (Fig. 1e). Although there are

alternative choices of ordering scheme, we chose Strahler ordering because it allows for a better assessment of self-similarity in an asymmetric branching structure[4, 67].

**Assessment of the self-thinning rule.** Using the average biomass to assess the self-thinning rule induces a bias towards small trees. This is because the size distribution contains a large number of very small trees. To avoid this bias, we follow a method proposed by Adler[45] and we use the effective number and the effective biomass. The effective number $\overline{N}$ is the inverse of the probability that two units of mass taken at random from all trees come from the same tree. The effective biomass $\overline{M}$ is the average biomass weighted by the biomass itself. These quantities are given by $\overline{N} = B_{\text{tot.}}^2/B_2$ and $\overline{M} = B_2/B_{\text{tot.}}$, with $B_{\text{tot.}}$ the total biomass and $B_2$ the second moment of the biomass distribution[45]. When all trees have the same size, $\overline{N}$ is the number of trees and $\overline{M}$ their biomass.

**Principle of the AMT model.** Consider a regular fractal tree skeleton of fractal dimension $D$ and branching ratio $R_n$. Because of the definition of $D$, the length ratio is simply $R_l = R_n^{1/D}$. In this regular architecture, a branch of Strahler rank $k$ is fed by $N_k = R_n^{k-1}$ foliages situated above in the hierarchy and its length is $l_k = LR_l^{k-1}$. Besides, the path length that connects the branch base to any descendant foliage is

$$\ell_k = l_k + l_{k-1} + \cdots + l_1 = L\frac{R_l^k - 1}{R_l - 1} \approx L\frac{R_l^k}{R_l - 1}. \qquad (5)$$

The bending moment at the base of this branch due to wind-induced loads in the foliages is

$$M_k = \frac{\alpha}{2}\rho U^2 S_{\text{fol.}} N_k \ell_k, \qquad (6)$$

with $\alpha = \alpha_1\alpha_2 < 1$ a geometrical parameter due to the fact that the distance between the branch base and the foliages is $\alpha_1$ times smaller than the path length, and that a branch is not necessarily orthogonal to the wind (the projection of the force is then $\alpha_2$ times smaller than in the orthogonal case).

It follows that the bending stress $\sigma_k$ at the base of the branch is (see section 'Wind-induced stress' in Methods section)

$$\frac{\sigma_k}{\sigma_0} = \frac{16\alpha}{\pi d_k^3} C_Y S_{\text{fol.}} N_k \ell_k. \qquad (7)$$

Then, since the safety factor $S$ of the different branches and different species does not vary substantially (generally $2.5 \lesssim S \lesssim 4$, Supplementary Fig. 6c), the branch diameter is given by

$$d_k^3 = \frac{16\alpha S^{\frac{3}{2}}}{\pi} C_Y S_{\text{fol.}} \frac{R_l L}{R_l - 1} R_n^{\frac{D+1}{D}(k-1)}, \qquad (8)$$

which means that the diameter ratio is $R_d = R_n^{(D+1)/3D}$ and $d_k = d_1 R_d^{k-1}$, with $d_1$ given by (8) for $k = 1$. For the particular case $D = 2.5$, one finds that $R_d = R_n^{7/15}$, $R_l = R_n^{2/5}$ and $R_d = R_l^{7/6}$ such that the aspect ratio $l_k/d_k$ is almost constant. In addition, the area varies as the square of the diameter, such that $R_a = R_n^{14/15}$, which means that the total cross-sectional area is almost constant across ranks.

Assuming now that the whole tree counts $K$ Strahler orders, the diameter of the axes is given by (8). The trunk is a special case because it is always vertical and orthogonal to the wind, and thus $\alpha_2^{-1/3}$ times larger than axes with random orientations

$$d_{\text{trunk}} = d_1' R_d^{K-1}, \quad \text{with } d_1' = \alpha_2^{-\frac{1}{3}}d_1 \qquad (9)$$

Then, the height of the tree is approximately

$$H \approx \ell_K = \frac{R_l L}{R_l - 1}\left(\frac{d_{\text{trunk}}}{d_1'}\right)^{\frac{3}{D+1}}, \qquad (10)$$

thus giving an allometric exponent for the tree height vs. trunk diameter, $\beta_H = \frac{3}{D+1}$. The total number of foliages is

$$N = R_n^{K-1} = \left(\frac{d_{\text{trunk}}}{d_1'}\right)^{\frac{3D}{D+1}}, \qquad (11)$$

such that the allometric exponent for $N$ is $\beta_N = \frac{3D}{D+1}$. The branch biomass volume is

$$B = \frac{\pi}{4}\left(d_K^2 l_K + R_n d_{K-1}^2 l_{K-1} + \cdots R_n^{K-1} d_1^2 l_1\right), \qquad (12)$$

$$= \frac{\pi d_1^2 L R_d^2 R_l}{4\left(R_d^2 R_l - R_n\right)}\left(\frac{d_{\text{trunk}}}{d_1'}\right)^{\frac{2D+5}{D+1}}, \qquad (13)$$

which gives an allometric exponent for the biomass $\beta_B = \frac{2D+5}{D+1}$. Finally, using (11) and (13), the exponent for the leaf biomass is found to be $\beta_{\text{ML}} = \frac{3D}{2D+5}$.

Taking $\alpha_1 = \frac{1}{2}$, $\alpha_2 = \frac{1}{2}$, $R_n = 3.5$, $D = 2.5$, $S = 3$, $S_{\text{fol.}} = \frac{1}{4}$, $C_Y = 2 \times 10^{-5}$, one finds $R_l = 1.65$, $R_d = 1.79$, which is coherent with independent measurements (Table 2 in the main text). One also finds $d_1' = 0.0552L$ and Eqs. (10), (11) and (13) yield

$$H \approx 37.1 L\left(\frac{d_{\text{trunk}}}{L}\right)^{0.857}, \qquad (14)$$

$$N \approx 814\left(\frac{d_{\text{trunk}}}{L}\right)^{2.14}, \qquad (15)$$

$$B \approx 33.9 L^3\left(\frac{d_{\text{trunk}}}{L}\right)^{2.857}. \qquad (16)$$

These allometric laws derived from the AMT model can be compared to the scaling relations found in the trees growing in the first-round virtual forests simulated with MECHATREE (Fig. 3b). Except for the number of foliages, which is somewhat over-predicted for large trees by this simple analytical model, there is a remarkable agreement with the data extracted from the simulated trees, thus proving that the arguments outlined above on mechanics and self-similarity capture well most of the mechanisms affecting allometry in MECHATREE (same is true for forests sown with only the two finalist species: Supplementary Fig. 4b).

To go further, one can try to compare the allometric scalings of the AMT model to the data measured empirically. To do so, we assume a segment length $L = 10$ cm, a dry wood density $\rho_{\text{wood}} = 613$ kg m$^{-3}$ and the mass of one foliage $m_{\text{fol.}} = 1.8$ g. Taking these values and using Eqs. (10), (11) and (13), the following relations are obtained

$$H \approx 26.7\, d_{\text{trunk}}^{0.857}, \qquad (17)$$

$$M_L \approx 202\, d_{\text{trunk}}^{2.14}, \qquad (18)$$

$$M_L \approx 0.124 M_S^{0.75}, \qquad (19)$$

where $M_S$ is the dry mass of stems and SI units have been used (metres for lengths and kilograms for weights). These relations show excellent agreement with allometric relations found in the literature (Fig. 6).

**Sensitivity analysis on the allometric laws.** We want to assess how the laws of tree allometry depend on the model parameters. To perform this sensitivity analysis, we have modified the Cauchy number $C_Y$, the maintenance thickness $e$, the volume produced by foliages $V_{\text{prod.}}$ and the foliage transparency $\alpha_{\text{fol.}}$, one-factor-at-a-time, around the reference values given in Table 1. Each allometric law can be written as $\alpha_X z^{\beta_X}$, where $\alpha_X$ is the intercept and $\beta_X$ is the exponent of the allometric relation. Both of these constants a priori depend on the model parameters. To characterise this dependency, we estimate the relative variation of each allometric constant for relative variation of each parameter independently, in other words the sensitivity

$$s_{\beta_X, p} = \left(\frac{p}{\beta_X}\right)_{\text{ref.}}\left(\frac{\partial\beta_X}{\partial p}\right)_{\text{ref.}}, \qquad (20)$$

where $p$ is one of the model parameter (with this definition, the sensitivities $s_{a,b}$ are always dimensionless).

To estimate the sensitivities, defined by Eq. (20), we have run 16 forest simulations during 3000 yrs, for each values of the model parameters. The macroscopic characteristics of each tree in these simulations (height, biomass, trunk diameter, etc.) are then extracted and used to perform a standard major axis regression similar to what has been done for the first-round forests (Fig. 3b). This is repeated for 6 different values of each of the 4 model parameters, resulting in a total of 384 simulations (taking ~1500 CPU-hours). The allometric constants calculated from these simulations are plotted in the Supplementary Fig. 11. The sensitivities are then estimated through the slope of the linear regression fit of these plots (Supplementary Table 2). Examining the values of the sensitivities in Supplementary Table 2, it appears that the allometric exponents $\beta_X$ do not vary significantly with the model parameters. There is only one exception: when the maintenance thickness is increased, or the volume produced by foliages decreased, $\beta_H$ increases. This occurs because, in these cases, trees are much smaller on average (see last column of Supplementary Fig. 11). As it has already been discussed in the main text, the relation between the logarithms of the trunk diameter and the tree height is not linear but curvilinear (probably because of finite-size effects). It results that a fit made on small trees tend to overestimate $\beta_H$. It thus explains the sensitivities $s_{\beta_H, e}$ and $s_{\beta_H, V_{\text{prod.}}}$ observed.

For the sensitivities on the allometric intercepts, a similar conclusion can be drawn: the dependencies on $C_Y$ and $\alpha_{\text{fol.}}$ can be interpreted with the AMT model (see below), and there is no significant dependency on $e$ and $V_{\text{prod.}}$, with the

exception of $\alpha_N$, for which the sensitivities are about $\pm 0.5$. One explanation for this specific behaviour may be that again when increasing $e$ or decreasing $V_{prod}$, less photosynthates are available for primary growth and thus trees are much smaller. There is also a selection of a lower safety against wind loads and a higher fractal dimension. All these effects concur to increase the number of leaves for a given trunk diameter and, as a consequence, increase $\alpha_N$, but further research is clearly needed to fully understand this effect.

From the analytic model for tree allometry (the AMT model), sensitivities can also be calculated (Supplementary Table 3). To do so, we have assumed that fractal dimension $D$ depends linearly on the foliage transparency $\alpha_{fol}$: $D = 2 + \alpha_{fol}$, as it is suggested from the parametric analyses (Supplementary Methods and Supplementary Fig. 12). For allometric intercepts, agreement with the numerical simulation is good: signs and orders of magnitude of the sensitivities are correctly recovered both for $C_Y$ and for $\alpha_{fol}$. For the allometric exponents, agreement is excellent, showing that the simple analytic AMT model correctly incorporates the principal mechanisms at play in setting the allometric laws in MECHATREE. It also shows that the hypothesis of linear dependence of fractal dimension on foliage transparency ($D = 2 + \alpha_{fol}$) is coherent with the sensitivities observed.

Supplementary Table 4 shows how fractal dimension, average tree size and safety are sensitive to the model parameters. It should be noted that the tree species used to compute these sensitivities have only evolved during 3000 yrs. On such a short period, only rudimentary selection has had enough time to act. Yet, safety against wind loads depend significantly on the model parameters. Our interpretation of this dependence is the following. As expected, trees can grow larger on average when more photosynthates can be allocated to primary growth (either when $C_Y$ or $e$ decreases, or when $V_{prod}$ or $\alpha_{fol}$ increases). However, trees that grow larger also take more time to grow. For them, a higher safety against wind loads may be necessary to survive very rare wind events. For these large trees, the fractal dimension also tends to be smaller because of finite-size effects. These arguments explain the signs of all sensitivities in Supplementary Table 4, except one: $s_{D,\alpha_{fol}}$. For larger transparencies of foliages, the fractal dimension tends to increase. The explanation for this particular dependence is given above: for opaque foliages, we expect a fractal dimension $D = 2$ and for fully transparent foliages, we expect volume-filling $D = 3$.

**Data availability**. The data sets generated during and/or analysed during the current study are available from the corresponding author on reasonable request.

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

## Acknowledgements

We kindly thanks Anne Atlan, Benjamin Audit, Eric Badel, Hugues Chaté, Yves Couder, Catherine Coutand, Thierry Fourcaud, Xavier Leoncini and Agnès Schermann, for stimulating and helpful discussions. This work was granted access to the HPC resources of Aix-Marseille Université, funded by the project Equip@Meso (ANR-10-EQPX-29-01) of the programme 'Investissements d'Avenir' supervised by the ANR.

## Author contributions

All authors conceived the study. C.E. wrote the code, carried out the simulations and wrote the first draft. All authors contributed to the revisions of the manuscript.

## Additional information

**Competing interests:** The authors declare no competing financial interests.

