## [Peer Review File · Nature Communications]

Reviewers' comments:

Reviewer #1 (Remarks to the Author):

Eloy et al. present a detailed and complex 3-d simulation of a growing tree subject to various rules associated with light interception, biomechanics of wind loading, and competition with neighboring trees. They develop this simulation model to address the question - what is the best allocation and growth strategy of trees? But more importantly, does this 'optimal strategy' yield tree architecture and allometry consistent with observed branching relationships and allometric scaling laws?

Eloy et al. model carbon allocation using formal neural networks. The resulting trees are simulated via beautiful 3-d rendering of realistic tree architectures. Eloy et al. argue that their approach allows for an agnostic and flexible modelling of complex physiological regulations. Their simulation platform 'mechatree' appears to allow one to simulate entire forests over long periods of time. The authors even suggest that this model may be able to replace the forest simulation models such as SORTIE.

Eloy et al make several conclusions and key statements from their simulations. First, they claim that their results "suggests a rethinking of tree allometry, whose origins should be sought in both mechanics and hydraulics." The implication is that the work of WBE (the analytical theory that Eloy et al. contrast their simulated findings with) does not capture the origin of tree allometry. Second, they claim that that self-similarity emerges from the complex rules of how trees branch and allocate given the demands of wind, biomechanics, and competition. Indeed they claim that self-similar branching and allometric properties of the 'optimal branching' simulated tree that emerged from 'mechatree' is in close to the measurements on existing tree species.

There are several elements to the approach and results that are appealing - clear representation of input rules that can influence plant growth and architecture, using competition and a genetic algorithm to allow branching architecture to evolve in differing environments, beautiful simulation graphics and movies, and connection with a wealth of scaling literature and fractal studies. Indeed, the connection between rules of branching to the structure and performance of whole phenotypes and forests is compelling. Nonetheless, there are several aspects of the current manuscript that were distracting and need to be addressed.

I have several reservations with the manuscript that question their central conclusions.

First, this really is a complicated simulation. Numerous parameters are used as inputs to simulate plant growth and competition. As a result, it is difficult to assess the importance of the various inputs and generality of the results. As a result, I am not fully convinced that the emergent 'optimal tree' is indeed optimal over all possible conditions. Specifically, the authors need to convince the reader that the behavior of the simulation has been fully vetted. Is the scaling behavior that emerges from the simulation due to initial conditions or

due to parameter level settings? For example, how sensitive are allometric relationships and self-similar ratios to variation to the numerous input factors including light, variation in wind, island size (perimeter ratio), leaf and terminal twig size, disturbance frequency, response to competition etc.? Several additional parameters are listed in Table 1. I would find it remarkable that the 'optimal tree' that emerges is the same across different environmental and biological conditions. No mention of a sensitivity analysis is given in the main text.

Second, the scholarly basis for justification of this model and the importance of the conclusions is set up as a straw man. This is not only distracting but limits the contribution to the literature. Eloy et al. argue that other models and approaches for the origin of allometric scaling have been either mechanical or hydraulic. They conclude that their approach is best because it includes both hydraulics and mechanics. But this is not correct. The work of WBE (specifically West et al. 1999 Nature but also more recently Savage et al. 2010 PNAS) is not just a hydraulic approach but instead includes both hydraulic and mechanical inputs and considerations. Eloy et al. state that from the WBE model "the observed allometric scalings result from the minimisation of hydraulic losses when water is transported". This is not entirely correct. The same allometric predictions also can emerge from including just space filling and biomechanical constraints on the scaling of branch radii and lengths. This is shown in West et al. 1999 as well as discussed in Savage et al. 2010 PNAS.

Third, I do not agree with how the authors have tested their simulation results with the theoretical predictions of WBE. Some of the key results of their simulation model are listed in Table III. Here the authors compare the fitted scaling relationships from their simulation with the analytical predictions of WBE. They claim that their results better fit the data. However, these are not accurate comparisons between the two models. As WBE have discussed (see WBE 1997 Science, 1999 Nature but also Enquist et al. 2007 Nature; Savage et al. 2009 PLoS Computational Biology) the WBE model actually predicts that several of these allometric scaling relationships are curvilinear (especially for the youngest trees); especially for the exponent Beta H. Its not surprising that the fitted power function exponent deviates because the data show curvilinearity in the smallest trees. The authors own simulation results also show this curvilinearity (Fig. 5a). Lastly, no formal statistical test is given comparing their simulation results with empirical data. At minimum confidence intervals are needed. The authors have also apparently missed some key literature here as well.

Lastly, I am not clear just how this simulation platform differs from other simulation platforms out there modeling the 3-D construction of trees or the origin of ecological scaling relationships (size distribution, crown packing etc.) . As the authors point out there are several other simulations available. What is missing is any scholarly contrasting with other simulation approaches. The work of Prusinkiewicz and colleagues appear to be quite similar to the approach of modeling branching geometry as presented by Eloy et al. However, no formal statistical comparison of these approaches is given. Also, the work of Hammond and Niklas is also quite similar in approach to the work presented here. Hammond and Niklas

present a simulation model that shows how the allometry of individuals then ramifies to influence the size and packing of individuals as well as the importance of competition on the evolution of optimal scaling exponents. However, again, no formal comparison is given. As a result, I cannot evaluate the novel contributions of the work of Eloy et al.

Other comments -

Branch death - it was not clear to me just how branches are shed and lost. A potentially novel aspect of this simulation model is the dynamic ontogenetic development of the plant branching network. Branches not only grow but they sometimes are lost. I had difficulty following through in their model exactly the rules governing just when branches are shed and lost. Is this due to carbon starvation?

Size distributions - Eloy et al. test the ecological predictions of their simulation model by assessing the relationship between average plant biomass and average population density. This relationship known as the self-thinning law has a long history in ecology. However, I am concerned that this relationship is not a strong test of their model. As discussed by White et al. 2007 TREE - there are statistical and methodological issues associated with taking the average. A stronger test is to not take an average but instead to assess the raw size distribution or size spectra.

Optimal branching architectures - I was surprised that no formal discussion is given to the work of Smith et al. (see Smith DD, Sperry JS, Enquist BJ, Savage VM, McCulloh KA, Bentley LP. Deviation from symmetrically self-similar branching in trees predicts altered hydraulics, mechanics, light interception and metabolic scaling. *New Phytologist*. 2014 Jan 1;201(1):217-29.). Smith et al. combine both analytical and simulation approaches to argue for an optimal branching allometry.

Awkward phrasing - there are several statements made throughout the paper that come off as incorrectly stated, teleological, or partially thought through. For example, statements like "The inputs are the amount of mobilisable biomass contained in the reserve and the number of foliages in the tree. The first and second outputs are the proportions of biomass allocated to new segments ($0 \leq P_{seg} \leq 1$) and to seeds" are biologically incorrect as biomass is not allocated or even mobilized - carbon, water, and nutrients are. Further, "The total volume requested by a [branch] segment is the sum of the volume needed to achieve a certain safety against wind loads and a maintenance volume calculated as . . ." Use of the terms 'requested' and 'needed' are teleological. Also, "If the biomass produced by a foliage exceeds the total amount requested by segments situated below, each segment receives its share and the leftover is stored in the reserve." Again, biomass is not 'requested' nor is biomass exchanged.

Also, in Figure 1, Biologically there is no such thing as some genal underground 'reserve' for plants. Yes, there are some storage that occurs but reproduction and growth tends to be largely fueled by local photosynthate production along closeby branches. I understand that some degree of abstraction is needed but for all of the discussion that this simulation model

accurately represents the physiology of plant allocation this is a glaring example of theoretical simplicity that negates any statement of biological realism of allocation.

Reviewer #2 (Remarks to the Author):

This paper is impressive because it develops a well-reasoned and sophisticated plant model that includes growth, branch dieback, wind, competition, and a variety of branching architectures. This is all accomplished in a thorough way. Intriguingly, the author reproduces many famous allometric scaling results both at the tree and forest level. Consequently, I think there is much to learn from this model, including via future iterations with water limitation, hydraulics, etc or as a means to study other questions about islands, evolution, and environmental stress on plants and forests. Moreover, the paper is well written, contains informative and clear figures, and both simulations and math appear to be well conceived, understood, and conducted. All of these reasons are evidence this will be an impactful paper.

I think the authors can improve a few aspects of this exciting paper including how to frame the results in relation to previous allometric models, hydraulic limitations, explaining the methods of their simulation, and the basis of their self similarity for branch radii. I now detail these comments as a list.

1. Through the Introduction and Discussion, the paper sets up a false dichotomy between mechanical and hydraulic models. WBE papers, Savage et al, and Smith et al models (Smith DD, Sperry JS, Enquist BJ, Savage VM, McCulloh KA, Bentley LP (2014) Deviation from symmetric self-similarity in trees predicts altered hydraulics, mechanics, light interception and metabolic scaling. *New Phytologist* 201: 217-229) all include both mechanical and hydraulic constraints. The mechanical constraints typically determine the scaling ratios for lengths of branches, and the hydraulics typically determine the scaling ratios for the radii of branches and xylem. Therefore, this part of the paper should be reformulated to say many models include both and both can be used to describe self similar aspects, depending on which ratios one is concerned with. Also, what about xylem scaling and embolism, which potentially combine mechanical and hydraulic constraints and are not in this model but could be important.

2. In this paper and model, the authors effectively assume unlimited water (largely removing hydraulic constraints and selection pressures), symmetric growth of branching, and other ideal conditions of their own. So it is not just WBE etc that are ideal cases. This could be more clearly stated.

3. The authors also chooses radius as constant proportion of length which means if the simulation produces self similarity for one aspect you will likely get it for the other by piggyback. So mechanical processes leading to self similarity in length might then lead to radius self similarity that just tag alongs by this initial choice made by the authors and thus be simply a correlation and unrelated to real biological growth or principles that lead to self similarity in radii scaling, which is hydraulic in WBE, Savage et al, and Smith et al.

In that way part of the self similarity may be because of model setup and hydraulics may be needed to obtain it for real systems.

4. The authors use Strahler labeling or ordering for generation number, whereas most of the other models listed above define generation number based on number of branching generations from heart. As a result, the scaling ratios, area conservation, and other properties may give different values of the exponents or different results based on which labeling for generation number is used and comparison with WBE and other models may be misleading. What are the authors results if they define generation number based on number of branchings from heart? Do their numbers change? My guess is that they will, and I wonder whether they are still as consistent with previous results. This is not to say that Strahler ordering is wrong but that one must be consistent in testing models to use generations numbers as defined in the model. It is possible Strahler is better for lengths and number of branching junctions is better for radii.

5. The authors give an interpretation of the simulation as a neural network. It is fine to use neural network programming but obviously plants don't have neurons, so could the authors choose a language in which the algorithmic choices and steps map in a more obvious to plant evolution? I think this would be helpful for the reader in understanding the algorithm.

6. Why are there 31 genes in the simulation? How was that number determined? The authors argue they have few parameters, which at one level seems justified, but couldn't these numbers of genes and neuronal levels, etc, be interpreted as a type of parameter, and if so, this is actually a large number of parameters for conducting the simulations and model. If this interpretation is incorrect about a larger number of parameters, the authors should explain why.

7. I believe some root systems have been measured and have been shown to exhibit area-preserving branching. The root architecture is not exposed to the same stress from wind as the branches aboveground, so how would the authors interpret area-preserving for root systems? Would it be mechanical or hydraulic or both and would it arise for the same or different reasons in roots and branches?

8. From the results in the Table comparing simulation results to model results such as WBE, it seems like WBE does as good or better on most predictions with a relatively simple framework. In the tables, it would be good to provide 95% confidence intervals to see if the simulations and the WBE model are equally consistent.

Reviewer #3 (Remarks to the Author):

The manuscript: Eloy et al. : Wind loads and competition for light sculpt trees into self-similar structures presents an interesting study that demonstrates how few simply formulated but essential features of tree development bring out realistic structure under competition for light and the selection pressure of wind induced stress when the evolution of the structural features is allowed. As such the model produces the outcome that has been

suggested using evolutionary arguments for necessary mechanical stability of trees but this model shows nicely how such an outcome could be achieved through reiterating growth of new tree segments, their subsequent thickening and pruning when allowing to act over several tree generations. For these reasons I think that the paper will be interesting for wide audience of Nature communication readers as similar demonstration of evolutionary development has not been shown before and therefore I can recommend its publication.

Having said the above, I think that in its present form the paper is too technically written for a non-specialist on plant development to appreciate the beauty of the work. As such, the model is well described but the authors should pay more effort to walk through the reader of the dynamic development how the structure evolves towards the presented outcome. The authors state the outcome "is actually the result of complex architecture reorganisations that occur during the lifetime of a tree through wind-induced pruning and light-dependent growth." However, it is not quite obvious how such reorganization takes place. Presumably trade-offs between light acquisition, segment mechanical stability and wind induced pruning are in action and it would be very interesting if the authors could demonstrate how they operate in this modelling exercise, since the outcome is very realistic. Demonstration of the sequence that lead to branch pruning vs not would be very helpful for the reader. Such treatment would also allow evaluation of the possible role of hydraulic limitation in the development of the final outcome as those principles have given an equally good structural explanation on tree structure using general evolutionary arguments of hydraulically efficient structures. From these results it would seem feasible that certain structural features have to be favorable both from mechanical and hydraulic point of view and it would be very interesting if these features could be identified in the evolution of the structure. It would be very interesting indeed to include some discussion if including both the features would have actually increased the speed of evolution as presumably double requirements for the performance of structure would have made non-optimal features more rapidly less favorable. Of course, this model is based on mechanical arguments but the authors have no doubt an idea where the hydraulic arguments could enter the scheme. Discussing that would open up the last sentence of the discussion: "A more constructive approach would be to consider tree allometry in the light of both mechanical and hydraulic arguments" and would enforce the conclusions of the work.

As more specific points, it would help the readers to appreciate the obtained resemblance to real trees if the author would open up what does variation in the compared allometric parameters and fractal dimension and how big a deviation is a big difference. There is some sensitivity analysis of the used parameters but it could be more comprehensive. I was particularly left wondering about the significance of the used mortality criteria as they seemed quite arbitrarily chosen.

Overall, I found the paper well written and clearly presented and referring to previous literature adequately but there were some technical problems. I agree with the authors that the model needs to be described in the detail they have chosen but I can hardly see it a part of the results. Also, perhaps somewhat shorter presentation of the model in the bulk text and more thorough description in the methods and supplementary materials would have left more room for the discussion, which I find too short as compared with the width of

the work.

Based on above, I recommend the publication of the manuscript but it would require a complete rewrite to make it more accessible for a general audience.

RESPONSE TO REVIEWERS

REVIEWER #1

Eloy et al. present a detailed and complex 3-d simulation of a growing tree subject to various rules associated with light interception, biomechanics of wind loading, and competition with neighboring trees. They develop this simulation model to address the question - what is the best allocation and growth strategy of trees? But more importantly, does this 'optimal strategy' yield tree architecture and allometry consistent with observed branching relationships and allometric scaling laws?

Eloy et al. model carbon allocation using formal neural networks. The resulting trees are simulated via beautiful 3-d rendering of realistic tree architectures. Eloy et al. argue that their approach allows for an agnostic and flexible modelling of complex physiological regulations. Their simulation platform 'MECHATREE' appears to allow one to simulate entire forests over long periods of time. The authors even suggest that this model may be able to replace the forest simulation models such as SORTIE.

Eloy et al make several conclusions and key statements from their simulations. First, they claim that their results "suggests a rethinking of tree allometry, whose origins should be sought in both mechanics and hydraulics." The implication is that the work of WBE (the analytical theory that Eloy et al. contrast their simulated findings with) does not capture the origin of tree allometry. Second, they claim that that self-similarity emerges from the complex rules of how trees branch and allocate given the demands of wind, biomechanics, and competition. Indeed they claim that self-similar branching and allometric properties of the 'optimal branching' simulated tree that emerged from 'MECHATREE' is in close to the measurements on existing tree species.

There are several elements to the approach and results that are appealing - clear representation of input rules that can influence plant growth and architecture, using competition and a genetic algorithm to allow branching architecture to evolve in differing environments, beautiful simulation graphics and movies, and connection with a wealth of scaling literature and fractal studies. Indeed, the connection between rules of branching to the structure and performance of whole phenotypes and forests is compelling.

Nonetheless, there are several aspects of the current manuscript that were distracting and need to be addressed. I have several reservations with the manuscript that question their central conclusions.

We thank the reviewer for his/her fair and thorough evaluation of our work. During the revision process, we have tried to address the issues raised. We believe that the manuscript is now clearer and stronger. Below is a point-by-point response to all comments. Changes in the manuscript and in the Supplementary Information have been coloured in red for convenience.

MAIN COMMENTS

QUESTION 1

First, this really is a complicated simulation. Numerous parameters are used as inputs to simulate plant growth and competition. As a result, it is difficult to assess the importance of the various inputs and generality of the results. As a result, I am not fully convinced that the emergent 'optimal tree' is indeed optimal over all possible conditions. Specifically, the authors need to convince the reader that the behavior of the simulation has been fully vetted. Is the scaling behavior that emerges from the simulation due to initial conditions or due to parameter level settings? For example, how sensitive are allometric relationships and self-similar ratios to variation to the numerous input factors including light, variation in wind, island size (perimeter ratio), leaf and terminal twig size, disturbance frequency, response to competition etc.? Several

additional parameters are listed in Table 1. I would find it remarkable that the 'optimal tree' that emerges is the same across different environmental and biological conditions. No mention of a sensitivity analysis is given in the main text.

The numerical simulations may seem complex, but the model has been designed with simplicity in mind. We have tried to have as few parameters as possible: foliage diameter, foliage transparency, Cauchy number, probability of gene mutations, etc. (Table I). The other factors affecting the growth are not parameters of the simulations but the outputs of the “genes” of each species. Because MECHATREE is including some (simplified) modelling of the Evolution, the “allelic” values of these “genes” (and therefore the value of the factors representing their output) may vary through mutation and their abundance in the population may vary due to selective pressure (fitness) and/or genetic drift. To our knowledge, MECHATREE is the first attempt to build a functional-structural model where most factors affecting branching geometry, carbon allocation, and growth are not fitted to match existing species but are able to evolve when species compete in a simplified environment.

That being said, we agree with the referee, it is important to assess how the final results may be affected by the different parameters of the simulation. With our current computer equipment (8 Intel i7 cores running in parallel), running a tournament such as the one illustrated in the Supplementary Figure 1 takes about 4 weeks. It would thus be very long to perform a real sensitivity analysis, i.e. run different tournaments, varying one-by-one all parameters and comparing the final fittest species. Even if we had enough computing resources to do so, the comparison would be very delicate since this is not an optimization calculation. The final fittest species, like for any evolutionary process, is dependent on the history, and somehow the role of chance cannot be completely ruled out.

To assess the influence of the simulation parameters on the self-similar characteristics of the fittest species, we have run parametric analyses on the fittest species (cf. sub-section “Parametric analyses” in the section “Results”, and Supplementary Fig. 5). In these analyses, the growth of the fittest species in an environment without any competitors is simulated by systematically changing the parameters one-by-one. The main result of these analyses is that only the foliage transparency has an influence on the branching ratio, the length ratio, and the fractal dimension. The other parameters affect the growth speed, the final size, but do not seem to affect the self-similar properties. This is further supported by the results we obtained using a preliminary version of the MECHATREE model, in which the foliages were fully opaque, the forest diameter was only $100L$ and the biomass produced by a foliage was $(5 V_0 l)$. With these parameters a complete tournament had been run (not shown in the present manuscript). The fittest species that emerged from this competition shared many characteristics with the fittest species arising from our present version of the model: similar $3/2$ self-thinning law, similar allometric exponents, similar growth strategy (safety against wind loads S approximately equal to 3, similar photosensitivity of the primary growth). The major difference was the fractal dimension that was around the value $D=2$. This is consistent with that was found with the fittest species of our current version of the model when the transparency of the foliages is set to zero (cf. Supplementary Fig. 5i).

To address the referee’s comment and to clarify these points, the section “Parametric analyses” has been entirely revised. During this revision process, we have tried to emphasize the points discussed above:

“In the present simulations, the fittest species is identified through an evolutionary tournament method. This approach is inherently stochastic and would produce slightly different results if it were run several times. As a consequence, tournaments would be difficult to use to perform a sensitivity analysis. In addition, running a tournament is computationally costly (about 3,000 CPU hours). To assess the sensitivity of the results to the model parameters, we turn to another method and perform parametric analyses instead.”

In these parametric analyses, we simulate during 200 yrs the growth of an isolated tree belonging to the fittest species. The reference case corresponds to values given in Table I and we compare this case to other runs with different values of the Cauchy number C_V , the maintenance thickness e , the volume of biomass produced $V_{prod.}$, and the foliage transparency $\alpha_{fol.}$ (Supplementary Fig. 5). The main result is that, although trees grow to a larger size when C_V or e is decreased, or when $V_{prod.}$ is increased, the fractal dimension remains in the interval $2.2 \leq D \leq 2.5$. However, the foliage transparency strongly affects the fractal dimension: $D \approx 2.1$ when foliage is fully opaque, and $D \approx 2.8$ when they are almost transparent (Supplementary Figs. 5h–i).

The dependence of the fractal dimension on the foliage transparency can be interpreted as follows. The outer surface covered by foliage clusters generally has a dimension around 2 (it can be slightly larger if it has some fractal roughness). When foliage clusters are opaque, foliage inside this surface does not intercept any light and will eventually be shed because the branches supporting them do not have the resources to ensure maintenance costs. Since foliage and the structure supporting them have generally the same dimension, we expect $D \approx 2$ for opaque foliages. This contrasts with the case of fully transparent foliages, where the structure is expected to be volume-filling (i.e. $D = 3$). Owing to the central role of chlorophyll in both photosynthesis and leaf transmittance properties, $V_{prod.}$ and $\alpha_{fol.}$ are likely to be negatively correlated. Whether there is an optimal transparency remains, however, an open question."

One of the main issues here is whether the foliage transparency (transmittance) is linked to chlorophyll content and photosynthetic capacity. Chlorophyll is surely an important aspect, but there are other factors of leaf transmittance (e.g. carotenoid, leaf thickness, scattering by air-water menisci, etc.), as well as other factors of photosynthetic capacity (rubisco, other components of the electronic chain, etc.). Surprisingly we could not find direct studies on the relations between the interspecific genetic variations of leaf transmittance, chlorophyll content, and photosynthetic capacity across a large sample of species. Poorter *et al.* (2000) reported data on 13 species from the cloud forest. They evidenced a relation between species-specific absorptance versus chlorophyll concentration per unit area (Fig 4a) but it was curvilinear and not very tight. Moreover, the "foliage transparency" is also related to the gap fraction in the foliage, i.e. to the leaf area density, to the angular distribution of leaf and to their degree of clumping (Godin and Sinoquet, 2005; Duursma *et al.* 2012). Again we could not find studies about the relationship between these traits and photosynthetic capacity, but one may speculate that these traits add new degrees of freedom in the relation between foliage transparency and photosynthate production. Therefore, the safest assumption so far is to assume that they can vary independently.

QUESTION 2

Second, the scholarly basis for justification of this model and the importance of the conclusions is set up as a straw man. This is not only distracting but limits the contribution to the literature. Eloy et al. argue that other models and approaches for the origin of allometric scaling have been either mechanical or hydraulic. They conclude that their approach is best because it includes both hydraulics and mechanics. But this is not correct. The work of WBE (specifically West et al. 1999 Nature but also more recently Savage et al. 2010 PNAS) is not just a hydraulic approach but instead includes both hydraulic and mechanical inputs and considerations. Eloy et al. state that from the WBE model "the observed allometric scalings result from the minimisation of hydraulic losses when water is transported". This is not entirely correct. The same allometric predictions also can emerge from including just space filling and biomechanical constraints on the scaling of branch radii and lengths. This is shown in West et al. 1999 as well as discussed in Savage et al. 2010 PNAS.

We thank the reviewer for drawing our attention to this point. Indeed, there are two versions of the WBE model: in a preliminary version (West, Brown, Enquist, *Science* 1997), the arguments are mainly hydraulic, while in a later version (West, Brown, Enquist, *Nature*, 1999; Savage *et al. PNAS* 2010) it is a combination of mechanical and hydraulic inputs. We have carefully read these papers again, and, as a result, we have revised the 4th paragraph of the introduction as follows.

“Among the hydraulic approaches, the pipe model [10], or the initial version of the West, Brown, and Enquist (WBE) model [19] have been highly influential. In these approaches, a tree is modelled as a fractal assembly of sap-conducting pipes. In its current version however, the WBE model also includes some mechanical principles [20, 21]: trees are modelled as volume filling self-similar networks following the principle of elastic similarity (i.e. branch radii and lengths are related by a power law such that the deflection of the branch tip under self-weight is proportional to the branch length, a relation that also meets a constant safety factor versus elastic buckling for upright axes) [3]. With these hypotheses, several allometric relations between trunk radius, tree height, stem biomass, and leaf biomass can be deduced without explicit reference to hydraulics. Within this mechanically-based structure, the authors claim that the minimisation of hydraulic losses yields allometric scalings for fluid velocity, hydraulic conductance, pressure gradient, etc.”

In addition, we added a point-by-point comparison between the WBE model and the analytical AMT model we propose (Supplementary Discussion, section 1.3). This led us to redo the calculations of the WBE model to predict not only the allometric exponents but also the allometric constants in order to perform a complete assessment (see Moulia and Fournier 1997). These results have been compared to empirical data found in the literature (Fig. 6). Both models (WBE and the proposed AMT model) appear to give similar results, both in very good agreement with the empirical data.

The main difference between the WBE and the AMT models comes from the hypotheses made. The hypotheses of volume filling and elastic similarity made in the WBE model seem somewhat less plausible than the fractal dimension around 2.5 and constant safety against wind loads that we use in MECHATREE. Indeed volume filling by foliage is usually partial (Zeide and Pfeifer 1991), and, from what we know from the plant biomechanics since the release of the elastic-similarity model in the mid 1970's, wind is by far a more challenging and selective mechanical load than buckling under self-weight in most environments except for the very thin suppressed understorey trees and –maybe– the emergent trees of exceptionally tall stature, i.e. tree champions (Moulia and Fournier 1997, Gardiner *et al.* 2016, King 1996). Moreover in our case the value of the fractal dimension and mechanical safety are in fact *emergent* properties of MECHATREE (see Supplementary Discussion, section 1.3). But anyhow the empirical data available at the moment do not allow to refute any of these approaches, as it is now stated in the Discussion:

“The WBE model is an analytical model based on the assumptions of self-similarity, volume filling, and elastic similarity. The allometric relations found with the WBE model are very similar to the results of the proposed AMT model (Table III, Fig. 6, and Supplementary Discussion, section 1.3). It is thus impossible to refute either approach based on allometry only. One way to achieve falsification would be to have more data on the fractal dimension, which should be possible with the development of Lidar-based technologies [65], or to directly assess the mechanistic bases of the processes involved, e.g. through wind mechanosensing [66].

In the WBE model, mechanics is implemented by enforcing an elastically-similar allometric relation between axis lengths and radii [3, 20, 21] (Supplementary Discussion, section 1.3). It is also assumed that the “metabolic performance of the tree” (e.g its photosynthesis) can be estimated from the water flux in the xylem through a constant water-use efficiency [67]. In MECHATREE, the mechanical stresses and the species-specific thigmomorphogenetic response

are calculated for each segment every year. Contrary to the WBE model, self-similarity and allometry are not imposed, but emerge from thigmomorphogenesis and competition for light.

The importance of hydraulics (water use efficiency, embolism) and mechanics (wind hazards, self-weight) may vary between species habitats and plant stages. In some cases, hydraulic performance may be a major selective pressure, whereas in others wind mechanical safety will be dominant. Yet these different situations may not be identified in broad range allometric data, since both selective pressures yield similar allometries. We may however speculate that the presence of both selective pressures can increase the speed of natural selection. We believe that further insights could be gathered by integrating the mechanically-based model proposed in the present paper with the hydraulic arguments of the WBE model to have a better understanding of both mechanics and hydraulics."

QUESTION 3

Third, I do not agree with how the authors have tested their simulation results with the theoretical predictions of WBE. Some of the key results of their simulation model are listed in Table III. Here the authors compare the fitted scaling relationships from their simulation with the analytical predictions of WBE. They claim that their results better fit the data. However, these are not accurate comparisons between the two models. As WBE have discussed (see WBE 1997 Science, 1999 Nature but also Enquist et al. 2007 Nature; Savage et al. 2009 PLoS Computational Biology) the WBE model actually predicts that several of these allometric scaling relationships are curvilinear (especially for the youngest trees); especially for the exponent β_H . It is not surprising that the fitted power function exponent deviates because the data show curvilinearity in the smallest trees. The authors own simulation results also show this curvilinearity (Fig. 5a). Lastly, no formal statistical test is given comparing their simulation results with empirical data. At minimum confidence intervals are needed. The authors have also apparently missed some key literature here as well.

We agree with the referee, our simulations and our analytical model do not do a better job at predicting allometric scalings than the WBE model. It has never been our intention to imply that our model better fits the data but just to show that it is an alternative to WBE that does as well as WBE. We have rephrased several portions of the manuscript to make this point clearer. For instance, this is what we now write in the Discussion:

"The allometric relations found with the WBE model are very similar to the results of the proposed AMT model (Table III, Fig. 6, and Supplementary Discussion, section 1.3). It is thus impossible to refute either approach based on allometry only."

Although the WBE model and our analytical model predict equally well the allometric scalings, we do believe that our mechanical hypotheses are resting on firmer ground than the mechanical hypotheses of the WBE model as it is discussed above (response to question 2).

We agree with the reviewer that some allometric relationships are curvilinear for the youngest or smallest trees. Finite size and finite branching order lead to nonlinear allometric scalings and this is likely why the exponent β_H is not properly defined. In response to this comment, we have revised the section "Tree allometry" as follows:

"In the literature, the allometric exponent for tree height has been measured to be in the interval $0.54 \leq \beta_H \leq 0.89$ [47–49]. The exponent $\beta_H = 2/3$ has also been predicted based on arguments of elastic similarity [3]. However, some authors argued that the relation between the logarithms of tree height and trunk diameter is not linear but curvilinear because of the finite size and the finite number of Strahler orders in the structure [10, 50, 51]. The same trend

is observed in our data: β_H tends to decrease with d_{trunk} (Fig. 3b). An additional reason for curvilinearity could be that young trees are not self-similar yet. As it shall be seen below, architecture reorganizations through photo-sensitive growth and wind-induced pruning take time.”

We should add that both the WBE and the AMT models suffer the same weakness: they are not suited to model architectures that are not self-similar, such as young trees or plants with a very small number of Strahler orders. For these trees or plants, it is a surprise that the same allometric scalings hold as it was already written in the previous version of the manuscript:

“Note also that agreement is excellent on a large interval of trunk diameters or stem dry mass (Fig. 6). This is rather surprising because the model should be limited to trees that are large enough to have a properly defined fractal dimension and small enough such that gravity is negligible.”

As suggested by the referee, confidence intervals have been given for the empirical data of Table III. Unfortunately, it is difficult to provide an in-depth statistical analysis of the merits and drawbacks of each model (WBE, SERA, and AMT). Since these models are not constructed as fits of empirical data with adjustable parameters, it is meaning-less to use a criterion such as the AIC, Akaike Information Criterion (Burham and Anderson, 2002), or any other similar method.

Finally, we added the references proposed by the referee to complete the literature on allometric scalings and allometric models.

QUESTION 4

Lastly, I am not clear just how this simulation platform differs from other simulation platforms out there modeling the 3-D construction of trees or the origin of ecological scaling relationships (size distribution, crown packing etc.). As the authors point out there are several other simulations available. What is missing is any scholarly contrasting with other simulation approaches. The work of Prusinkiewicz and colleagues appear to be quite similar to the approach of modeling branching geometry as presented by Eloy et al. However, no formal statistical comparison of these approaches is given. Also, the work of Hammond and Niklas is also quite similar in approach to the work presented here. Hammond and Niklas present a simulation model that shows how the allometry of individuals then ramifies to influence the size and packing of individuals as well as the importance of competition on the evolution of optimal scaling exponents. However, again, no formal comparison is given. As a result, I cannot evaluate the novel contributions of the work of Eloy et al.

First, we agree with the reviewer: MECHATREE shares similarities with some works of the Prusinkiewicz’s group. This is why this approach has been discussed in the Introduction:

“Models exploiting the recursive characteristics of tree architectures [31] have also been developed by using the formal grammar of L-systems [32]. Throughout the years, these models have evolved towards a better account of biological mechanisms [33]. In particular, the self-organising processes involved during growth through competition for light and interactions with the environment have been addressed [34].”

However these models do not incorporate mechanosensitive responses as we do by modifying radial growth to respond dynamically to the wind stresses, but sometimes integrate biomechanical growth inputs through the use of a priori allometries. More importantly, to our knowledge, these models have never been used to predict quantitatively allometric laws. This is why it would be difficult to perform a formal statistical comparison with these approaches, as suggested by the reviewer.

Second, the reviewer mentions the allometric scalings predicted by SERA, the individual-based model developed by Hammond and Niklas. In the SERA model, the allometry at the tree level is specified through a set of species-specific parameters. When this model is run, the authors show that allometric scalings emerge at the population level (height vs. trunk diameter, leaf biomass vs. stem biomass, etc.). In the revised version of our manuscript, we now compare these allometric scalings with the other models in Table III. In addition, we have added the following paragraph to the Discussion:

“The SERA model is an individual-based model with species-specific parameters describing how an individual tree grows when it competes for space and light [63]. From this model emerge allometric relations at the population level (Table III). In that sense, it bears some similarity with the present study. There are two major differences between MECHATREE and SERA however: MECHATREE does not involve species-specific parameters other than those selected by the simulated evolution in the model; and the modelling units are the branch segments, which allows us to address the origin of self-similarity in individual trees.”

OTHER COMMENTS

Branch death - it was not clear to me just how branches are shed and lost. A potentially novel aspect of this simulation model is the dynamic ontogenetic development of the plant branching network. Branches not only grow but they sometimes are lost. I had difficulty following through in their model exactly the rules governing just when branches are shed and lost. Is this due to carbon starvation?

The reviewer is correct: branch dies usually under carbon starvation. An equivalent of carbon starvation occurs in our model when a branch cannot ensure its maintenance costs, resulting in a mechanical weakening of the branch. When carbon starvation occurs, the weakened branch falls and is definitely shed through mechanical wind-induced pruning. We have added the following paragraph at the end of the section “Growth” to clarify this point:

“Within this model, branch fall is provoked by mechanical loading, and this can occur along two different scenarios: either an extreme wind event occurs and branches can fracture with a probability described by a Weibull distribution (see Methods), or the foliage sources cannot provide enough photosynthates to ensure the maintenance costs of a given branch and this branch will weaken to the point where it will fall down whatever the level of mechanical load.”

Size distributions - Eloy et al. test the ecological predictions of their simulation model by assessing the relationship between average plant biomass and average population density. This relationship known as the self-thinning law has a long history in ecology. However, I am concerned that this relationship is not a strong test of their model. As discussed by White et al. 2007 TREE - there are statistical and methodological issues associated with taking the average. A stronger test is to not take an average but instead to assess the raw size distribution or size spectra.

We thank the reviewer for drawing our attention to this point. To assess the self-thinning rule, one cannot simply take the average biomass over all trees otherwise the results would be biased towards very small trees. We were already aware of this possible artefact and this is why we use an “effective biomass” and an “effective number”, following a method proposed by Adler (1996). To make this point clearer, we have revised the section “Self-thinning”:

“The $-3/2$ self-thinning law for plant populations states that the average biomass of plants decreases as $n^{-3/2}$, with n the plant density [43]. This relation can be assessed by computing how the “effective number” and “effective biomass” vary with time [44] (see Methods).”

We also added a section “Assessment of the self-thinning rule” in Methods:

“Using the average biomass to assess the self-thinning rule induces a bias towards small trees. This is because the size distribution contains a large number of very small trees. To avoid this bias, we follow a method proposed by Adler [44] and we use the effective number and the effective biomass. The effective number N is the inverse of the probability that two units of mass taken at random from all trees come from the same tree. The effective biomass M is the average biomass weighted by the biomass itself. These quantities are given by $N = B_{\text{tot.}}^2 / B_2$ and $M = B_2 / B_{\text{tot.}}$, with $B_{\text{tot.}}$ the total biomass and B_2 the second moment of the biomass distribution [44]. When all trees have the same size, N is the number of trees and M their biomass.”

We certainly agree that size distributions are a stronger test than (weighted) averages. Unfortunately, there is no equivalent of the $-3/2$ self-thinning law for size distributions in a young forest. However, in an old-growth forest at equilibrium, the size distribution is expected to follow a universal d^{-2} distribution (Enquist and Niklas, *Nature*, 2001, White et al., *Trends Ecol. Evol.*, 2007). This allometric scaling has been tested and we added a new figure to show that the forests simulated by MECHATREE indeed follow this scaling law (Supplementary Fig. 3). To refer to this new figure, we have added the following text in the section “Tree allometry”:

“In addition, the distribution of trunk diameters in the forests composed of the finalist species scales as d^{-2} (Supplementary Fig. 3). This scaling has been observed in many forest communities and has been predicted by the theory of metabolic ecology [45, 46].”

Optimal branching architectures - I was surprised that no formal discussion is given to the work of Smith et al. (see Smith DD, Sperry JS, Enquist BJ, Savage VM, McCulloh KA, Bentley LP. Deviation from symmetrically self-similar branching in trees predicts altered hydraulics, mechanics, light interception and metabolic scaling. *New Phytologist*. 2014 Jan 1;201(1):217-29.). Smith et al. combine both analytical and simulation approaches to argue for an optimal branching allometry.

Following the reviewer’s suggestion, we have added a reference to Smith et al. (2014) and we now discuss the asymmetry of self-similar branching at the end of the section “Self-similar ratios”:

“The self-similar architecture pictured in Fig. 4 is an asymmetric binary tree (indicated by $R_n > 2$), which calls for specific models of fractal geometry [52]. It has been suggested that there may be an optimal asymmetry that maximises biomass production per biomass investment [53]. In MECHATREE, this asymmetry is not imposed, it spontaneously emerges from the model dynamics through architecture reorganisations. Whether this spontaneous asymmetry is optimal or not remains to be determined.”

Awkward phrasing - there are several statements made throughout the paper that come off as incorrectly stated, teleological, or partially thought through. For example, statements like "The inputs are the amount of mobilisable biomass contained in the reserve and the number of foliages in the tree. The first and second outputs are the proportions of biomass allocated to new segments ($0 \leq P_{\text{seg}} \leq 1$) and to seeds" are biologically incorrect as biomass is not allocated or even mobilized - carbon, water, and nutrients are. Further, "The total volume requested by a [branch] segment is the sum of the volume needed to achieve a certain safety against wind loads and a maintenance volume calculated as . . ." Use of the terms 'requested' and 'needed' are teleological. Also, "If the biomass produced by a foliage exceeds the total amount requested by segments situated below, each segment receives its share and the leftover is stored in the reserve." Again, biomass is not 'requested' nor is biomass exchanged.

We agree with the reviewer that some statements in the first version of our manuscript involved some easing with a current, physiologically consistent vocabulary. However, choosing the proper level of disciplinary vocabulary is made difficult in a multidisciplinary context, such as the readership of *Nature Communications*,

which reflects the different backgrounds of the authors (physics, biomechanics, ecophysiology, and forest ecology). In our initial submission, we tried to avoid disciplinary phrasing that may seem jargonic for other disciplines. Although we were aware that biomass is not “allocated” or “demanded” (one of us is a specialist in assimilate partitioning/storage and phloem biophysics/physiology), we thought it was appropriate because it has been widely used in the plant modelling literature (e.g. Google Scholar finds over 33,500 articles containing the phrase “biomass allocation”). We should add that our aim was not to precisely describe the physiological processes at play in plants, but to describe how MECHATREE works and is implemented.

We have complied with the reviewer’s suggestion, trying to find phrasing that could be both physiologically accurate and yet fairly easy to understand for people with other disciplinary backgrounds. As suggested by the reviewer, when referring to the allocation processes we now use “carbon” or “photosynthates” instead of “biomass”, and “sink strength” instead of “demand” or “request”.

Also, in Figure 1, Biologically there is no such thing as some general underground ‘reserve’ for plants. Yes, there are some storage that occurs but reproduction and growth tends to be largely fueled by local photosynthate production along closeby branches. I understand that some degree of abstraction is needed but for all of the discussion that this simulation model accurately represents the physiology of plant allocation this is a glaring example of theoretical simplicity that negates any statement of biological realism of allocation.

We understand that the reviewer may have been upset with the terminological issues already discussed above. The point of the relative involvement of the daily local photosynthates pool and of the different reserves has been indeed the matter of many investigations in plant ecophysiology. We thought we were clear about the reason why we were not accurately locating the assimilate pools involved when we stated in the previous version of the manuscript that:

“The reserve, whose exact location is not important, stores assimilates from the current year that will be mobilised the following year to support primary and reproductive growth.”

To precise our point, we have added the following sentence just after the above:

“Reserves have been included because of their significant impact on tree capacity to recover from major disturbances like strong wind damages as simulated in the model.”

However, the reviewer is right, this deserves a more cautious biological explanation. Reserves may indeed not be of major significance in regular, “easy” conditions, *i.e.* in the absence of any “major disturbances” such as fire, severe drought, flooding, storm, insect defoliation, *etc.* However, reserves have long been shown to have a major impact on tree growth, reproduction, and even survival, when such “major disturbances” do occur (Wargo, 1981; Carrol *et al.*, 1983; Gregory and Wargo, 1986). Due to their long lifespan, chances that trees may face such disturbances are not negligible, all the less so as they grow older. As the occurrence and consequences of one such disaster, namely strong wind damage, is explicitly modelled and discussed in this paper as a major determinant of tree longevity and forest evolution, reserves had to be explicitly included, albeit in a very simple way with no specific reserve storage location.

REVIEWER #2

This paper is impressive because it develops a well-reasoned and sophisticated plant model that includes growth, branch dieback, wind, competition, and a variety of branching architectures. This is all accomplished in a thorough way. Intriguingly, the authors reproduce many famous allometric scaling results both at the tree and forest level. Consequently, I think there is much to learn from this model, including via future iterations with water limitation, hydraulics, etc or as a means to study other questions about islands, evolution, and environmental stress on plants and forests. Moreover, the paper is well written, contains informative and clear figures, and both simulations and math appear to be well conceived, understood, and conducted. All of these reasons are evidence this will be an impactful paper.

I think the authors can improve a few aspects of this exciting paper including how to frame the results in relation to previous allometric models, hydraulic limitations, explaining the methods of their simulation, and the basis of their self-similarity for branch radii. I now detail these comments as a list.

Thank you for these very positive comments; we of course appreciated this feedback on our work!. In revising the manuscript, we have tried to address the concerns raised. We believe that the manuscript is now clearer and stronger. Below is a point-by-point response to all comments. Changes in the manuscript and in the Supplementary Information have been coloured in red for convenience.

QUESTION 1

Through the Introduction and Discussion, the paper sets up a false dichotomy between mechanical and hydraulic models. WBE papers, Savage et al, and Smith et al models (Smith DD, Sperry JS, Enquist BJ, Savage VM, McCulloh KA, Bentley LP (2014) Deviation from symmetric self-similarity in trees predicts altered hydraulics, mechanics, light interception and metabolic scaling. *New Phytologist* 201: 217-229) all include both mechanical and hydraulic constraints. The mechanical constraints typically determine the scaling ratios for lengths of branches, and the hydraulics typically determine the scaling ratios for the radii of branches and xylem. Therefore, this part of the paper should be reformulated to say many models include both and both can be used to describe self similar aspects, depending on which ratios one is concerned with. Also, what about xylem scaling and embolism, which potentially combine mechanical and hydraulic constraints and are not in this model but could be important.

Following the reviewer's suggestion, we have revised the Introduction and the Discussion to clarify the (mechanical) hypotheses of the WBE model. There are in fact two versions of the WBE model: in a preliminary version (West, Brown, Enquist, *Science* 1997), the arguments are mainly hydraulic, while in a later version (West, Brown, Enquist, *Nature*, 1999; Savage *et al.* *PNAS* 2010) it is a combination of mechanical and hydraulic inputs. We have carefully read these papers again, and, as a result, we have revised the 4th paragraph of the introduction as follows.

"Among the hydraulic approaches, the pipe model [10], or the initial version of the West, Brown, and Enquist (WBE) model [19] have been highly influential. In these approaches, a tree is modelled as a fractal assembly of sap-conducting pipes. In its current version however, the WBE model also includes some mechanical principles [20, 21]: trees are modelled as volume filling self-similar networks following the principle of elastic similarity (i.e. branch radii and lengths are related by a power law such that the deflection of the branch tip under self-weight is proportional to the branch length, a relation that also meets a constant safety factor versus elastic buckling for upright axes) [3]. With these hypotheses, several allometric relations between trunk radius, tree height, stem biomass, and leaf biomass can be deduced without explicit reference to hydraulics. Within this mechanically-based structure, the authors claim that the minimisation of hydraulic losses yields allometric scalings for fluid velocity, hydraulic conductance, pressure gradient, etc."

In addition, we added a point-by-point comparison between the WBE model and the analytical AMT model we propose (Supplementary Discussion, section 1.3). This led us to redo the calculations of the WBE model to predict not only the allometric exponents but also the allometric constants in order to perform a complete assessment (see Moulia and Fournier 1997). These results have been compared to empirical data found in the literature (Fig. 6). Both models (WBE and the proposed AMT model) appear to give similar results, both in very good agreement with the empirical data.

The main difference between the WBE and the AMT models comes from the hypotheses made. The hypotheses of volume filling and elastic similarity made in the WBE model seem somewhat less plausible than the fractal dimension around 2.5 and constant safety against wind loads that we use in MECHATREE. Indeed volume filling by foliage is usually partial (Zeide and Pfeifer 1991), and wind is by far a more challenging and selective mechanical load than buckling under self-weight in most environments except for the very thin suppressed understorey trees and the emergent trees of exceptionally tall stature, i.e. tree champions (Moulia and Fournier 1997, Gardiner et al. 2016, King 1996). Moreover in our case the value of the fractal dimension and mechanical safety are in fact not a priori stated but *emergent* properties of MECHATREE (see Supplementary Discussion, section 1.3). But anyhow the empirical data available at the moment do not allow to refute any of these approaches, as it is now stated in the Discussion:

“The WBE model is an analytical model based on the assumptions of self-similarity, volume filling, and elastic similarity. The allometric relations found with the WBE model are very similar to the results of the proposed AMT model (Table III, Fig. 6, and Supplementary Discussion, section 1.3). It is thus impossible to refute either approach based on allometry only. One way to achieve falsification would be to have more data on the fractal dimension, which should be possible with the development of Lidar-based technologies [65], or to directly assess the mechanistic bases of the processes involved, e.g. through wind mechanosensing [66].”

As for the reference to embolism, we agree with the reviewer. It would indeed be a constraint on hydraulics that may have been overlooked in the original WBE model, but has been partially addressed by Savage et al. (2010). However, in the present paper, we are mainly concerned by mechanics (wind-induced loads) and competition for light, for which the explicit description of hydraulic circulation is not necessary. Introducing the risk of embolism into MECHATREE will be a nice avenue for future works, as it is now suggested at the end of the Discussion:

“The importance of hydraulics (water use efficiency, embolism) and mechanics (wind hazards, self-weight) may vary between species habitats and plant stages. In some cases, hydraulic performance may be a major selective pressure, whereas in others wind mechanical safety will be dominant. Yet these different situations may not be identified in broad range allometric data, since both selective pressures yield similar allometries. We may however speculate that the presence of both selective pressures can increase the speed of natural selection. We believe that further insights could be gathered by integrating the mechanically-based model proposed in the present paper with the hydraulic arguments of the WBE model to have a better understanding of both mechanics and hydraulics.”

QUESTION 2

In this paper and model, the authors effectively assume unlimited water (largely removing hydraulic constraints and selection pressures), symmetric growth of branching, and other ideal conditions of their own. So it is not just WBE etc that are ideal cases. This could be more clearly stated.

We agree with the referee, MECHATREE is a parcimonious model and thus involves a lot of approximations and simplifications. We have revised the first paragraph of the section “Results” to make this point clearer:

“Our aim is to mimic the main characteristics of an angiosperm-like phenotypic set [37]. In building MECHATREE, several simplifications have been made, with the goal of keeping the model parsimonious and manageable. In particular, we have neglected the selective pressure exerted by hydraulics through the cost of transport and embolism. This is by no means because hydraulics is not important for trees, but we wanted to check how the predictions of a model based on light competition and mechanical response to wind compare with hydraulically-based models.”

However, we do not think that MECHATREE and the WBE model belong to the same class of models. This is the reason why we have constructed a mechanically-based *analytical* model, the AMT model (for Analytical MechaTree), which belongs to the same class of models as the WBE model and can be directly compared to it. In the AMT model though, the assumptions made are based on the *emergent* properties of the simulations done with MECHATREE, as it is now explained in the section “The AMT model: an analytical model of allometry”:

“Genotypes that survive the initial 3,000 yrs of simulation exhibit allometric relations close to the ones observed in nature (Table III). Interestingly, these relations can be recovered with a simple mechanically-based analytical model that we shall call the AMT (Analytical MechaTree) model.

To build this analytical model, we use the three emergent results of MECHATREE:

- i. Trees are self-similar*
- ii. Fractal dimension is $D \approx 2.5$*
- iii. Trees have a constant safety against wind loads”*

QUESTION 3

The authors also choose radius as constant proportion of length which means if the simulation produces self similarity for one aspect you will likely get it for the other by piggyback. So mechanical processes leading to self similarity in length might then lead to radius self similarity that just tag alongs by this initial choice made by the authors and thus be simply a correlation and unrelated to real biological growth or principles that lead to self similarity in radii scaling, which is hydraulic in WBE, Savage et al, and Smith et al. In that way part of the self similarity may be because of model setup and hydraulics may be needed to obtain it for real systems.

Maybe we have not been clear enough in our manuscript, but branch radii are not chosen in constant proportion to branch lengths and that is indeed a major difference between our model and other ones. Branch lengths results from the architecture reorganisations. Initially, branch lengths are all equal to the segment length L . As the tree grows, lateral segments will eventually be pruned, and several segments will “assemble” end-to-end to produce longer branches. In parallel, branch radii increase through a thigmomorphogenetic response to wind-induced stresses, that has no direct dependency on branch length.

This aspect is now emphasized in the Introduction:

“This popular concept of optimal mechanical design is related to the well-established process of thigmomorphogenesis, which is the plant response to mechanical signals [16, 17]. The ecological significance of thigmomorphogenetic acclimation has long been recognised [18], even though it has never been implemented in ecological forest models so far.”

As a result, in the trees selected by MECHATREE evolutionary tournaments, the proportion between branch lengths and branch radii is not constant. It varies between 9 and 25 for the fittest species, as it is already mentioned in the section “Self-similar ratios”. The scatter in Fig. 5d also shows that branch radii are not constant for a given branch rank, or for a given average path-length to the distal ends.

This is a major difference with the WBE model, or other models implementing mechanical inputs only indirectly through an allometric relation between branch lengths and branch radii. In these models, the argument of elastic similarity is used (linked to an argument of elastic buckling, but we have argued in Supplementary Discussion, section 1.3, that buckling is not applicable to non-vertical branches). In MECHATREE, we do not use a global allometric relation to implement “mechanical aspects” as it is now more clearly stated in the Discussion:

“In the WBE model, mechanics is implemented by enforcing an elastically-similar allometric relation between axis lengths and radii [3, 20, 21] (Supplementary Discussion, section 1.3). It is also assumed that the “metabolic performance of the tree” (e.g. its photosynthesis) can be estimated from the water flux in the xylem through a constant water-use efficiency [67]. In MECHATREE, the mechanical stresses and the species-specific thigmomorphogenetic response are calculated for each segment every year. Contrary to the WBE model self-similarity and allometry are not imposed, but emerge from thigmomorphogenesis and competition for light.”

QUESTION 4

The authors use Strahler labeling or ordering for generation number, whereas most of the other models listed above define generation number based on number of branching generations from heart. As a result, the scaling ratios, area conservation, and other properties may give different values of the exponents or different results based on which labeling for generation number is used and comparison with WBE and other models may be misleading. What are the authors results if they define generation number based on number of branchings from heart? Do their numbers change? My guess is that they will, and I wonder whether they are still as consistent with previous results. This is not to say that Strahler ordering is wrong but that one must be consistent in testing models to use generations numbers as defined in the model. It is possible Strahler is better for lengths and number of branching junctions is better for radii.

We agree with the reviewer: changing the ordering scheme would change the results on the self-similar ratios. This question is certainly important, and it has been discussed in the past in the context of river networks (Strahler, *Geol. Soc. Am.*, 1953; Horsfield 1972), or in the context of tree structures (McMahon & Kronauer, *J. Theor. Biol.*, 1976; Turcotte, Pelletier & Newman, *J. Theor. Biol.*, 1998).

For symmetric branching, the choice of a particular ordering scheme is not crucial and this is why the WBE model or other models with symmetric binary trees use different ordering schemes. For asymmetric branching however, the Strahler ordering scheme seems to be superior and this is why we used it. Here “superior” means that it allows for a linear relation between the logarithm of the quantities (number of branches, mean radius, mean length) and the order (as in Fig. 5b). We have revised the section “Strahler order” in Methods to clarify this point:

“Although there are alternative choices of ordering scheme, we chose Strahler ordering because it allows for a better assessment of self-similarity in an asymmetric branching structure [3, 52].”

QUESTION 5

The authors give an interpretation of the simulation as a neural network. It is fine to use neural network programming but obviously plants don't have neurons, so could the authors choose a language in which the algorithmic choices and steps map in a more obvious to plant evolution? I think this would be helpful for the reader in understanding the algorithm.

We have rephrased the section “Growth”, as suggested by the reviewer:

“Here, neural networks are used as tool that allows for an agnostic and flexible modelling of complex physiological regulations that do not involve actual neurons [38].”

QUESTION 6

Why are there 31 genes in the simulation? How was that number determined? The authors argue they have few parameters, which at one level seems justified, but couldn't these numbers of genes and neuronal levels, etc, be interpreted as a type of parameter, and if so, this is actually a large number of parameters for conducting the simulations and model. If this interpretation is incorrect about a larger number of parameters, the authors should explain why.

In response to this comment, we now explain why there are 31 genes in the section “Growth”:

“In the present case, there are 31 genes: 3 genes for the geometric angles of branching, and 18 and 10 genes for the neural coefficient of the primary- and secondary-growth neural networks respectively. These 31 genes are complemented with 3 “neutral marker genes” used for visualisation purposes.”

As the reviewer have remarked, we have tried to have as few parameters as possible: 10 parameters describing the tree (foliage diameter, foliage transparency, Cauchy number, probability of gene mutations, etc.) and 1 parameter describing the size of the forested island in which Evolution takes place (Table I). The other factors affecting the growth are not parameters of the simulations but genetic variables controlled by the genes of each species. To our knowledge, MECHATREE is the first attempt to build a functional-structural model where most factors affecting branching geometry, carbon allocation, growth, and mortality are not fitted to match existing species but are able to evolve when species compete in a simplified environment.

A possible drawback of having a large number of genes might be that it prevents natural selection to be achieved. However, this limitation is not relevant as long as the simulations are run for a sufficiently long time such that evolution has allowed the selection of the “fittest” genes. In our case, this is indeed what happens because simulations are run until there is only one or very few species remaining on the virtual island.

From the point of view of quantitative genetics now, it may seem that 31 genes is very little, compared to real genomes. However, the model only deals with a minimal amount of processes. Besides, our genes can be viewed as major Quantitative Trait Loci (QTLs) explaining most of the genetic variability in the characters, as used in advanced genetic-ecophysiological crop models (*e.g.* Reymond et al 2003).

QUESTION 7

I believe some root systems have been measured and have been shown to exhibit area-preserving branching. The root architecture is not exposed to the same stress from wind as the branches aboveground, so how would the authors interpret area-preserving for root systems? Would it be mechanical or hydraulic or both and would it arise for the same or different reasons in roots and branches?

The reviewer is probably referring to the paper of Oppelt *et al.* (2001), which is, to our knowledge, the only paper measuring area conservation in root systems. We do not have an explanation for this particular observation, although it may be linked to hydraulics. Note however that wind loads also have major direct and indirect thigmomorphogenetic effects on the growth and morphology of the root system (see Coutand 2010, for a review). In any case, MECHATREE has not been developed to address this issue. We built MECHATREE to investigate the development of aerial architectures and the possible consequences of light competition and wind effects. Area conservation is just one of the allometric relations that are tested to assess the credibility of the model, it is not the core of the paper.

We believe that studying root development and mechanical (and hydraulic) functioning falls out of the scope of the present manuscript. It would likely require a substantial amount of work and should be a study on its own

QUESTION 8

From the results in the Table comparing simulation results to model results such as WBE, it seems like WBE does as good or better on most predictions with a relatively simple framework. In the tables, it would be good to provide 95% confidence intervals to see if the simulations and the WBE model are equally consistent.

As requested by the referee, we have provided 95% confidence intervals for the empirical data in Table III.

As it is explained above (response to Question 1), we also added a detailed comparison between the results of the WBE model and our proposed analytical AMT model (Supplementary Discussion, section 1.3). In particular, we show that the complexity, the number of assumptions, and the resulting allometric relations are similar. The main difference between the WBE and the AMT models comes from the assumptions: in the AMT model these assumptions are based on emergent properties of MECHATREE, whereas in the WBE model they are hypotheses.

REVIEWER #3

The manuscript: Eloy et al. : “Wind loads and competition for light sculpt trees into self-similar structures” presents an interesting study that demonstrates how few simply formulated but essential features of tree development bring out realistic structure under competition for light and the selection pressure of wind induced stress when the evolution of the structural features is allowed. As such the model produces the outcome that has been suggested using evolutionary arguments for necessary mechanical stability of trees but this model shows nicely how such an outcome could be achieved through reiterating growth of new tree segments, their subsequent thickening and pruning when allowing to act over several tree generations. For these reasons I think that the paper will be interesting for wide audience of Nature communication readers as similar demonstration of evolutionary development has not been shown before and therefore I can recommend its publication.

Thank you for this positive evaluation of our work. Following your comments and the comments of the other reviewers, we have entirely revised the manuscript. We believe that it is now clearer and stronger. Below is a point-by-point response to all comments. Changes in the manuscript and in the Supplementary Information have been coloured in red for convenience.

QUESTION 1

Having said the above, I think that in its present form the paper is too technically written for a non-specialist on plant development to appreciate the beauty of the work. As such, the model is well described but the authors should pay more effort to walk through the reader of the dynamic development how the structure evolves towards the presented outcome. The authors state the outcome "is actually the result of complex architecture reorganisations that occur during the lifetime of a tree through wind-induced pruning and light-dependent growth." However, it is not quite obvious how such reorganization takes place. Presumably trade-offs between light acquisition, segment mechanical stability and wind induced pruning are in action and it would be very interesting if the authors could demonstrate how they operate in this modelling exercise, since the outcome is very realistic. Demonstration of the sequence that lead to branch pruning vs not would be very helpful for the reader. Such treatment would also allow evaluation of the possible role of hydraulic limitation in the development of the final outcome as those principles have given an equally good structural explanation on tree structure using general evolutionary arguments of hydraulically efficient structures. From these results it would seem feasible that certain structural features have to be favorable both from mechanical and hydraulic point of view and it would be very interesting if these features could be identified in the evolution of the structure. It would be very interesting indeed to include some discussion if including both the features would have actually increased the speed of evolution as presumably double requirements for the performance of structure would have made non-optimal features more rapidly less favorable. Of course, this model is based on mechanical arguments but the authors have no doubt an idea where the hydraulic arguments could enter the scheme. Discussing that would open up the last sentence of the discussion: "A more constructive approach would be to consider tree allometry in the light of both mechanical and hydraulic arguments" and would enforce the conclusions of the work.

To address the first point on “how such reorganization takes place” and on “the sequence that lead to branch pruning”, we have added the following paragraph at the end of the section “Growth”:

“Within this model, only wind-induced loading can lead to branch shedding, but this can occur along two different scenarios: either an extreme wind event occurs and branches can fracture with a probability described by a Weibull distribution (see Methods); or the foliage sources cannot provide enough photosynthates to ensure the maintenance costs of a given branch and this branch will weaken to the point where it will fall down whatever the level of mechanical load.”

As the reviewer suggests, there are likely to be “trade-offs between light acquisition, segment mechanical stability and wind induced pruning”. In MECHATREE these trade-offs can emerge because the development of the structure requires to fulfil functions (primary growth, light interception, stem secondary growth, mechanical safety), have costs, and compete for the photosynthates. Interestingly, through *in silico* evolution, there is a selection of certain traits that are common to all species that survived the initial round of the single-elimination tournament. In particular, these species share similar growth strategies and similar safety against wind-induced loads. We have revised the section “Tree allometry” to emphasize this point:

“These allometric scalings emerge because the trees selected through the single-elimination tournament share common characteristics. All have a similar safety factor against wind loads $S \approx 3$ (Supplementary Fig. 7c), and they are self-similar with a fractal dimension around $D \approx 2.5$ (Supplementary Fig. 7e).”

The second point of this question concerns “the possible role of hydraulic limitation in the development of the final outcome”. This is certainly an interesting point and we agree with the reviewer that in some cases the selective pressure for hydraulic and mechanical performance may lead to similar features (which presumably increase the selective pressure on these traits). It is however beyond the scope of the present paper to include an explicit description of hydraulics in MECHATREE as it is now more clearly stated in the section “Structural units”:

“Our aim is to mimic the main characteristics of an angiosperm-like phenotypic set [37]. In building MECHATREE, several simplifications have been made, with the goal of keeping the model parsimonious and manageable. In particular, we have neglected the selective pressure exerted by hydraulics through the cost of transport and embolism. This is by no means because hydraulics is not important for trees, but we wanted to check how the predictions of a model based on light competition and mechanical response to wind compare with hydraulically-based models.”

Nonetheless, hydraulics should be taken into account in future studies if one wants to fully understand the selective pressures on tree growth. We have revised the Discussion to clarify our point of view:

“The importance of hydraulics (water use efficiency, embolism) and mechanics (wind hazards, self-weight) may vary between species habitats and plant stages. In some cases, hydraulic performance may be a major selective pressure, whereas in others wind mechanical safety will be dominant. Yet these different situations may not be identified in broad range allometric data, since both selective pressures yield similar allometries. We may however speculate that the presence of both selective pressures can increase the speed of natural selection. We believe that further insights could be gathered by integrating the mechanically-based model proposed in the present paper with the hydraulic arguments of the WBE model to have a better understanding of both mechanics and hydraulics.”

Note also that we have added an entire section to discuss the similarities and difference between the WBE model and the analytical model we propose (Supplementary Discussion, section 1.3).

QUESTION 2

As more specific points, it would help the readers to appreciate the obtained resemblance to real trees if the author would open up what does variation in the compared allometric parameters and fractal dimension and how big a deviation is a big difference. There is some sensitivity analysis of the used parameters but it could be more comprehensive. I was particularly left wondering about the significance of the used mortality criteria as they seemed quite arbitrarily chosen.

We agree with the referee: it is important to assess how the final results may be affected by the different parameters of the simulation. Running a tournament such as the one illustrated in the Supplementary Figure 1 takes about 4 weeks. It would thus be very long to perform a real sensitivity analysis, i.e. run different tournaments, varying one-by-one all parameters and comparing the final fittest species. Even if we had enough computing resources to do so, the comparison would be very delicate since this is not an optimization calculation. The final fittest species, like for any evolutionary process, is dependent on the history, and somehow the role of chance cannot be completely ruled out.

To circumvent this problem and still give an idea of the sensitivity of the model output to parameters values, our strategy has been to assess the influence of the simulation parameters on the self-similar characteristics of the fittest species, we have run “Parametric analyses” (cf. sub-section “Parametric analyses” in the section “Results”, and Supplementary Fig. 5). In these analyses, the growth of the fittest species in an environment without any competitors is simulated by systematically changing the parameters one-by-one. The main result of these analyses is that only the foliage transparency has an influence on the branching ratio, the length ratio, and the fractal dimension. The other parameters affect the growth speed, the final size, but do not seem to affect the self-similar properties.

Since these parametric analyses are based on the simulated growth of a single tree without competitor, they are not suited to study the influence of the parameters specific to the forest community (Forest radius, death rules), or to the mutation/evolution process (mutation probability, mutation amplitude). It is therefore difficult to assess the sensitivity of the mortality criteria chosen, even if we agree that these criteria may appear arbitrary. The mortality criteria we retained are described in the section “Competition”:

“5. Death. A tree is removed if one of these two conditions is realised: (i) its age is larger than 1000 yrs; (ii) its age is larger than 6 yrs and its number of segments is less than 10.”

To address the referee’s comment and to clarify these points, the section “Parametric analyses” has been entirely revised. During this revision process, we have tried to emphasize the points discussed above:

“In the present simulations, the fittest species is identified through an evolutionary tournament method. This approach is inherently stochastic and would produce slightly different results if it were run several times. As a consequence, tournaments would be difficult to use to perform a sensitivity analysis. In addition, running a tournament is computationally costly (about 3,000 CPU hours). To assess the sensitivity of the results to the model parameters, we turn to another method and perform parametric analyses instead.

In these parametric analyses, we simulate during 200 yrs the growth of an isolated tree belonging to the fittest species. The reference case corresponds to values given in Table I and we compare this case to other runs with different values of the Cauchy number C_V , the maintenance thickness e , the volume of biomass produced V_{prod} , and the foliage transparency α_{fol} . (Supplementary Fig. 5). The main result is that, although trees grow to a larger size when C_V or e is decreased, or when V_{prod} is increased, the fractal dimension remains in the interval $2.2 \leq D \leq 2.5$. However, the foliage transparency strongly affects the fractal dimension: $D \approx 2.1$ when foliage is fully opaque, and $D \approx 2.8$ when they are almost transparent (Supplementary Figs. 5h–i).

The dependence of the fractal dimension on the foliage transparency can be interpreted as follows. The outer surface covered by foliage clusters generally has a dimension around 2 (it can be slightly larger if it has some fractal roughness). When foliage clusters are opaque, foliage inside this surface does not intercept any light and will eventually be shed because the branches supporting them do not have the resources to ensure maintenance costs. Since

foliages and the structure supporting them have generally the same dimension, we expect $D \approx 2$ for opaque foliages. This contrasts with the case of fully transparent foliages, where the structure is expected to be volume-filling (i.e. $D = 3$). Owing to the central role of chlorophyll in both photosynthesis and leaf transmittance properties, $V_{prod.}$ and $\alpha_{fol.}$ are likely to be negatively correlated. Whether there is an optimal transparency remains, however, an open question.”

QUESTION 3

Overall, I found the paper well written and clearly presented and referring to previous literature adequately but there were some technical problems. I agree with the authors that the model needs to be described in the detail they have chosen but I can hardly see it a part of the results. Also, perhaps somewhat shorter presentation of the model in the bulk text and more thorough description in the methods and supplementary materials would have left more room for the discussion, which I find too short as compared with the width of the work. Based on above, I recommend the publication of the manuscript but it would require a complete rewrite to make it more accessible for a general audience.

The manuscript has been largely remodelled an rewritten, with the goal of addressing the points raised by the three reviewers, while keeping the text accessible for a general audience (with a necessary trade-off between these two –sometimes conflicting– requirements!). Following the suggestion of the reviewer, we have postponed to Methods some technical aspects of the model. There are now 4 new sections in Methods: “Formal neural networks”, “Primary growth”, “Genome and mutation”, and “Assessment of the self-thinning rule”. In response to the comments of all reviewers, we have also expanded the descriptions of the Results (sections “Tree allometry”, “Self-similar ratios”, “Parametric analyses”) and the Discussion.

It should be noted however, that the name of the main sections (Results, Discussion, Methods) are imposed by the journal. This is why the description of the model appears in the Results, whereas –and we agree with the reviewer on this point– it would be more natural to have it in a separate section.

REFERENCES

- K. Burham, D. Anderson. Information theory and loglikelihood models: a basis for model selection and inference. *In: Model Selection and Multimodel Inference: A practical Information-Theoretic approach*. Berlin, Springer, 2002.
- J. E. Carroll and T. A. Tattar. Relationship of root starch to decline of sugar maple. *Plant Dis.*, 67:1347–1349, 1983.
- S. Coste, J.-C. Roggy, P. Imbert, C. Born, D. Bonal, and E. Dreyer. Leaf photosynthetic traits of 14 tropical rain forest species in relation to leaf nitrogen concentration and shade tolerance. *Tree Physiol.*, 25:1127–1137, 2005.
- C. Coutand. Mechanosensing and thigmomorphogenesis, a physiological and biomechanical point of view. *Plant Sci.*, 179:168–182, 2010.
- R. A. Duursma, D. S. Falster, F. Valladares, F. J. Sterck, R. W. Pearcy, C. H. Lusk, K. M. Sendall, M. Nordenstahl, N. C. Houter, B. J. Atwell, et al. Light interception efficiency explained by two simple variables: a test using a diversity of small-to medium-sized woody plants. *New Phytol.*, 193:397–408, 2012.
- B. Gardiner, P. Berry, and B. Moulia. Review: Wind impacts on plant growth, mechanics and damage. *Plant Sci.*, 245:94–118, 2016.
- C. Godin and H. Sinoquet. Functional–structural plant modelling. *New Phytol.*, 166:705–708, 2005.
- R. A. Gregory and P. M. Wargo. Timing of defoliation and its effect on bud development, starch reserves, and sap sugar concentration in sugar maple. *Can. J. For. Res.*, 16:10–17, 1986.
- D. A. King. Allometry and life history of tropical trees. *J. Trop. Ecol.*, 12:25–44, 1996.
- B. Moulia, C. Coutand, and J.-L. Julien. Mechanosensitive control of plant growth: bearing the load, sensing, transducing, and responding. *Front. Plant Sci.*, 6:52, 2015.
- B. Moulia and M. Fournier-Djimbi. Optimal mechanical design of plant stems: the models behind the allometric power laws. *In G. Jeronimidis and J. F. V. Vincent, editors, Proceedings of the Plant Biomechanics*. Reading Univ, 1997.
- Oppelt, W. Kurth, and D. Godbold. Topology, scaling relations and leonardo’s rule in root systems from african tree species. *Tree Physiology*, 21(2-3):117, 2001.
- L. Poorter, R. Kwant, R. Hernandez, E. Medina, and M. J. A. Werger. Leaf optical properties in venezuelan cloud forest trees. *Tree Physiol.*, 20:519–526, 2000.
- M. Reymond, B. Muller, A. Leonardi, A. Charcosset, and F. Tardieu. Combining quantitative trait loci analysis and an ecophysiological model to analyze the genetic variability of the responses of maize leaf growth to temperature and water deficit. *Plant Physiology*, 131:664–675, 2003.
- P. M. Wargo. Defoliation, dieback and mortality. In: "The gypsy moth. Research toward integrated pest management. Chapter 5. Effects of defoliation". *Tech. Bull. USDA*, (1584), 240-248., 1981.
- Zeide and P. Pfeifer. A method for estimation of fractal dimension of tree crowns. *Forest Sci.*, 37:1253–1265, 1991.

Reviewers' comments:

Reviewer #2 (Remarks to the Author):

As before, I think it is great work and contributions in terms of the simulation, the nice figures, the inclusion of wind stress, and the attempt to compare allometric results to other models. However, there are two major problems that were all raised by referees in the previous round that have still not been correctly or adequately addressed. For this round I focus on only those two major issues.

1. The referees pointed out that some sort of sensitivity analysis is needed for the study to assess the robustness and validity of the findings. The authors response seems to be that doing this is hard and that there is some fragility (meaning sensitivity/lack of robustness) based on randomness and repeated histories in the simulations. These responses are not vary satisfactory because the answers to these questions are important. With regard to the fragility of repeated histories, this is exactly why a sensitivity analysis and multiple runs need to be conducted. That is, to make sure these results aren't an unlikely outlier but instead are typical results for the principles and setup given by the authors. If they are not, the very foundation and basis of the paper is missing. The claim that doing a sensitivity analysis and repeating simulations takes a lot of time is certainly true. There are ways around this, using clusters or parallelizing, but I am sympathetic to this claim. Nevertheless, the authors do no one a service by trying to argue it doesn't matter or isn't crucial. It is crucial, and if the authors cannot do it right now, they should simply admit this as a shortcoming of the current paper and say that it needs to be done in future work or in a future paper and that the results here are intriguing but more work needs to be done to validate them and understand their robustness. For instance, understanding the effect of mortality more fully is important.

Perhaps as importantly is the finding that foliage transparency has a big effect on the scaling exponent. This does not seem to have much to do with wind stress or hydraulics, but rather competition for light. WBE and other models effectively assume light can go through the whole canopy so that you get volume filling, but if leaves are opaque, you just fill disks, not spheres. In this sense WBE and similar models are one end of the spectrum of this parameter. The authors actually state this clearly later in the paper, but it is already after much discussion of differences between mechanics and hydraulics in models that still gives an impression of that WBE or hydraulic or other models don't include such effects at all. Many of these points are in the paper, but it is presented in a confusing and not very coherent manner in which different parts of the paper seem to contradict each other in this revised form.

2. The comparisons with the WBE model and hydraulic models are still not correct. Here are several ways in which it is not correct. What is new about the authors' model is its ability to incorporate wind stress specifically along with other factors and to do so through large evolutionary and growth simulations with lots of trees, which is impressive, but they need to strive to be transparent about its shortcomings, the shortcomings of the present analysis, and the diversity, complexity, and advantages of other existing models. This all needs to be embraced in revision. The authors have done impressive work, but they need to be

transparent about its shortcomings and do a better job of putting it in the context of other work.

a. The authors talk about current models of WBE and include references to Savage et al, Smith et al, etc. Those models are not current versions of WBE. They may be WBE related or inspired but West and Brown were not part of the work, and the models differ in important ways.

b. In Table 1 there is a comparison with "WBE models", but which model is meant? As stated above there are several different models being referenced here with potential differences in results. And as said above, these aren't all WBE models?

c. The authors say in reference to WBE that "Its base is however a self-similar mechanical model and different allometric scaling laws can be derived without explicit reference to hydraulics." This statement is confusing and wrong at several levels. First, many predictions of WBE require both the mechanics and hydraulics, so it is not very easy to separate them, and for the predictions where it is possible, the authors should state exactly which ones and if those are the ones being compared to. Second, I thought the authors were referring to the WBE models as a hydraulic model to compare to, but now it seems they are saying hydraulics are not important for WBE. For instance, in the sentence: "We believe that further insights could be gathered by integrating the mechanically-based model proposed in the present paper with the hydraulic arguments of the WBE model to have a better understanding of both mechanics and hydraulics." suggests WBE is purely hydraulic. There is also the sentences, "The WBE model is an analytical model based on the assumptions of self-similarity, volume filling, and elastic similarity." which focuses only on the mechanics and not the hydraulics. So, is the claim that WBE is about mechanics or hydraulics or both. The answer is both, and for later non-WBE models (such as Savage et al and Smith et al) this is even more true, but the authors seem to alternate inconsistently between calling them just hydraulic or just mechanical. Third, the WBE model in any form predicts self similarity but does not assume self similarity, though a couple of places in the manuscript it is said to assume self similarity.

d. The authors keep talking about hydraulic models when in fact many of these models are a combination of hydraulics and mechanics. That is, they are not purely hydraulic, and this needs to be stated more clearly and explicitly when the term hydraulic is used by itself so much. The authors seem to not acknowledge that good models already exist that incorporate hydraulics and mechanics and have predictions that rely on both and yield different types of predictions.

3. There are a few typos such as this missing "be": "The dependence of the fractal dimension on the foliage transparency can be interpreted as follows." The manuscripts should be proofread carefully once more.

Reviewer #3 (Remarks to the Author):

I am happy with the authors responses to the concerns I had of the previous version of the ms and recommend its publication.

RESPONSE TO REVIEWER #2

As before, I think it is great work and contributions in terms of the simulation, the nice figures, the inclusion of wind stress, and the attempt to compare allometric results to other models.

Thank you again for this very positive evaluation of our work.

However, there are two major problems that were all raised by referees in the previous round that have still not been correctly or adequately addressed. For this round I focus on only those two major issues.

Please, find a point-by-point answer below. In revising the manuscript for the second time, we have made an effort to follow all the reviewer's suggestions. For practical purposes, changes in the manuscript and in the Supplementary Information have been coloured in blue.

QUESTION 1

The referees pointed out that some sort of sensitivity analysis is needed for the study to assess the robustness and validity of the findings. The authors response seems to be that doing this is hard and that there is some fragility (meaning sensitivity/lack of robustness) based on randomness and repeated histories in the simulations. These responses are not very satisfactory because the answers to these questions are important. With regard to the fragility of repeated histories, this is exactly why a sensitivity analysis and multiple runs need to be conducted. That is, to make sure these results aren't an unlikely outlier but instead are typical results for the principles and setup given by the authors. If they are not, the very foundation and basis of the paper is missing. The claim that doing a sensitivity analysis and repeating simulations takes a lot of time is certainly true. There are ways around this, using clusters or parallelizing, but I am sympathetic to this claim.

Nevertheless, the authors do no one a service by trying to argue it doesn't matter or isn't crucial. It is crucial, and if the authors cannot do it right now, they should simply admit this as a shortcoming of the current paper and say that it needs to be done in future work or in a future paper and that the results here are intriguing but more work needs to be done to validate them and understand their robustness. For instance, understanding the effect of mortality more fully is important.

We agree with the referee, testing the robustness of the results is important. In response to this comment, we have performed a sensitivity analysis. This analysis took several weeks of parallel computing and allowed to test that allometric exponents do not vary significantly when model parameters are varied. In other words, the allometric laws emerging from MechaTree are robust to variations of the model parameters. The results of this analysis are summarized in the main text (section "Sensitivity analysis"), and more thoroughly explained in the Supplementary Discussion (the new section 1.4, Tables II to IV, and Supplementary Fig. 8).

In addition, the self-similar characteristics of the other finalist species are now better described (new Supplementary Figs 6-7). This species had an entirely different evolution path than the fittest species and yet its self-similar characteristics are very similar. It thus seems that, in our simulations, the fitness landscape has a clear maximum and our artificial Evolution is not trapped into a non-representative local extremum. This can be viewed as a test of the robustness and repeatability of the selection despite the inherent stochasticity of evolution – or, at least, the simplified version of evolution used in the paper.

Perhaps as importantly is the finding that foliage transparency has a big effect on the scaling exponent. This does not seem to have much to do with wind stress or hydraulics, but rather competition for light.

Again, we agree with the referee: the relation between fractal dimension and foliage transparency is one of the important results of the paper and it is mainly related to light interception, not to mechanics or hydraulics. However, light interception alone is not enough: a set of mechanisms that can remodel tree architecture are

also needed (here wind-induced pruning, and growth strategy). This has been overlooked in the literature so far and we think that this is a new insight gathered through our approach.

However, as we stated in the previous round (see just below), robust experimental data are still needed to fully explore this aspect. In particular, we expected to find data showing that the volume of photosynthates produced by foliages, $V_{\text{prod.}}$, and transparency, $\alpha_{\text{fol.}}$, are negatively correlated. But apparently, there are no such data yet (and the constitution of such database on a large sample of species would require a very large amount of work). As we wrote in our answer to reviewer #1 in the previous round:

“One the main issue here is whether the foliage transparency (transmittance) is linked to chlorophyll content and photosynthetic capacity. Chlorophyll is surely an important aspect, but there are other factors of leaf transmittance (e.g. carotenoid, leaf thickness, scattering by air-water menisci, etc.), as well as other factors of photosynthetic capacity (rubisco, other components of the electronic chain, etc.). Surprisingly we could not find direct studies on the relations between the interspecific genetic variations of leaf transmittance, chlorophyll content, and photosynthetic capacity across a large sample of species. Poorter et al. (2000) reported data on 13 species from the cloud forest. They evidenced a relation between species-specific absorptance versus chlorophyll concentration per unit area (Fig 4a) but it was curvilinear and not very tight. Moreover, the “foliage transparency” is also related to the gap fraction in the foliage, i.e. to the leaf area density, to the angular distribution of leaf and to their degree of clumping (Godin and Sinoquet, 2005; Duursma et al. 2012). Again, we could not find studies about the relationship between these traits and photosynthetic capacity, but one may speculate that these traits add new degrees of freedom in the relation between foliage transparency and photosynthate production. Therefore, the safest assumption so far is to assume that they can vary independently.”

WBE and other models effectively assume light can go through the whole canopy so that you get volume filling, but if leaves are opaque, you just fill disks, not spheres. In this sense WBE and similar models are one end of the spectrum of this parameter. The authors actually state this clearly later in the paper, but it is already after much discussion of differences between mechanics and hydraulics in models that still gives an impression of that WBE or hydraulic or other models don't include such effects at all. Many of these points are in the paper, but it is presented in a confusing and not very coherent manner in which different parts of the paper seem to contradict each other in this revised form.

We partly agree with the first part of the reviewer's comment: concerning light interception, WBE and similar models are an extreme end of the spectrum.

In the original WBE model however, light interception is not related to the fractal dimension, but to tree height. Fractal dimension is simply assumed to be $D=3$ (volume filling) without any particular justification or any link to light interception.

“The model is based on a few general principles: (1) the branching network is volume filling; (2) the leaf and petiole size are invariant; (3) biomechanical constraints are uniform; and (4) energy dissipated in fluid flow is minimized.” — G. B. West, J. H. Brown, B. J. Enquist. 1999. A general model for the structure and allometry of plant vascular systems. Nature, 400, 66-667.

*“Competition for light has apparently led to a design that maximizes canopy height and simultaneously minimizes tapering of vascular tubes.” — *ibid.**

To the best of our knowledge, in other models, hydraulic or not, light interception and its relation to fractal dimension is not addressed either.

Within the WBE framework, the sensitivity to foliage transparency has not been assessed, because the physics of light interception and the possible dynamical remodelling of plant architecture are not taken into account. This is one of the reasons why we decided to build a dynamical structure-function model and why we included a light-interception module. Within MechaTree, we could indeed assess whether the outputs of the model were sensitive to foliage transparency.

In the second part of this comment, the reviewer writes that “different parts of the paper seem to contradict each other in this revised form”. We carefully read our manuscript focusing on the mention of the fractal dimension or transparency to understand why the reviewer felt that some parts were in contradiction with each other. Fractal dimension is first introduced in the section “Self-similar ratios”. At this point, there is just a definition and a description of the results without any attempt to link it to any cause. Then, in the section “Sensitivity analysis”, the fractal dimension is shown to depend on the foliage transparency through sensitivity and parametric analyses. In section “The AMT model”, there is a comparison with measured fractal dimensions reported in the literature, but again no mention of any cause. Finally, in the section “Discussion”, the results are summarized and the relation between the fractal dimension and the foliage transparency is mentioned again. Assuming that this is where the problems came from, we have revised this part in order to state more explicitly our viewpoint:

“Two important results can be deduced from MECHATREE. First, fractal dimension D mainly results from the consequences of competition for light on the architectural remodelling. As a result, D is strongly linked to foliage transparency: $D = 2.5$ for a transparency of $\alpha_{fol} = 0.5$. Second, for realistic probability of extreme wind events, the safety factor against wind loads is approximately constant ($S \approx 3$), which leads to area conservation.”

Hopefully this will clarify our point of view and address the reviewer’s concern.

QUESTION 2

The comparisons with the WBE model and hydraulic models are still not correct. Here are several ways in which it is not correct. What is new about the authors' model is its ability to incorporate wind stress specifically along with other factors and to do so through large evolutionary and growth simulations with lots of trees, which is impressive, but they need to strive to be transparent about its shortcomings, the shortcomings of the present analysis, and the diversity, complexity, and advantages of other existing models. This all needs to be embraced in revision. The authors have done impressive work, but they need to be transparent about its shortcomings and do a better job of putting it in the context of other work.

- a. The authors talk about current models of WBE and include references to Savage et al, Smith et al, etc. Those models are not current versions of WBE. They may be WBE related or inspired but West and Brown were not part of the work, and the models differ in important ways.**
- b. In Table 1 there is a comparison with "WBE models", but which model is meant? As stated above there are several different models being referenced here with potential differences in results. And as said above, these aren't all WBE models?**

We understand that this is a terminological problem. Actually in the manuscript, we only referred to Savage et al. (2010) as being a WBE model, because this was suggested by reviewer #1 in the previous round:

“The work of WBE (specifically West et al. 1999 Nature but also more recently Savage et al. 2010 PNAS) is not just a hydraulic approach but instead includes both hydraulic and mechanical inputs and considerations.” — Reviewer #1 (first round of reviews)

This made us believe that WBE could be used as a label for the whole set of models that have built upon the original WBE papers. To follow the reviewer’s suggestion, we now contrast the WBE model itself with the “WBE-related” models (e.g., “In its current version however, the WBE model [20], or related models [5, 21],

also include some mechanical principles” in the Introduction). In Table III, we now only refer to the original WBE model.

c. The authors say in reference to WBE that "Its base is however a self-similar mechanical model and different allometric scaling laws can be derived without explicit reference to hydraulics." This statement is confusing and wrong at several levels. First, many predictions of WBE require both the mechanics and hydraulics, so it is not very easy to separate them, and for the predictions where it is possible, the authors should state exactly which ones and if those are the ones being compared to. Second, I thought the authors were referring to the WBE models as a hydraulic model to compare to, but now it seems they are saying hydraulics are not important for WBE. For instance, in the sentence: "We believe that further insights could be gathered by integrating the mechanically-based model proposed in the present paper with the hydraulic arguments of the WBE model to have a better understanding of both mechanics and hydraulics." suggests WBE is purely hydraulic. There is also the sentences, "The WBE model is an analytical model based on the assumptions of self-similarity, volume filling, and elastic similarity." which focuses only on the mechanics and not the hydraulics. So, is the claim that WBE is about mechanics or hydraulics or both. The answer is both, and for later non-WBE models (such as Savage et al and Smith et al) this is even more true, but the authors seem to alternative inconsistently between calling them just hydraulic or just mechanical. Third, the WBE model in any form predicts self-similarity but does not assume self-similarity, though a couple of places in the manuscript it is said to assume self-similarity.

Here, we have to disagree with the reviewer along three lines.

First of all, as stated in the previous round, it is confusing to state that a model includes “mechanics” (meaning solid mechanics). There are many mechanical challenges that an erected land plant has to face. One is its interaction with wind (a fluid-solid interaction actually, see for example de Langre 2008; Gardiner, Berry & Mouliia 2016), leading to wind-induced bending stresses in the plant tissues. The second is Euler buckling and post-buckling bending under self-weight. The mechanics of the two processes differs broadly (one involves fracture due to a bending wind-loads, the other a bending instability due to compression). MechaTree includes a simulation of bending under wind-loads. In the WBE and WBE-related models, “mechanics” refers to buckling (at least for the orthotropic axes). But there is no explicit mechanical modelling of the buckling under self-weight, just the use of an allometric scaling for the relationship between total length and basal diameter (slenderness) and for the diameter as function of position (tapering). This allometric scaling has been interpreted by Thomas Mc Mahon (Science 1973) as resulting from the selection of a safety margin against buckling for erected axes, and for maximal lateral span for horizontal branches (elastic similarity). However, since the 1970’s the study of plant biomechanics has made huge progress. Recently, in a consensus statement by 18 authorities in the field of tree biomechanics and damages (Albrecht et al. 2016,), it has been stated that

“The conclusion reached [...] depends on the assumption of the interspecific scaling relationship $H \propto D^{2/3}$ (wherein H = height and D = trunk diameter), which emerges from the hypothesis of elastic self-similarity [6, 8]. However, elastic self-similarity is based on buckling under self-weight, which is almost unheard of in the natural world except in the case of extremely slender trees growing under unusual circumstances [9]. In addition, comparisons with broad datasets have shown that this scaling holds true only for trees of exceptionally tall stature, many of which emerge above the canopy of the surrounding forest (i.e., tree champions). For example, this scaling relationship does not hold for even-aged forest. [...] Indeed, the size, shape, and material properties of all plants including trees are responsive to wind-induced mechanical loads during their growth, a phenomenon called thigmomorphogenesis [11–13]. Thus, trees growing in sheltered conditions differ in height, trunk diameter, and wood mechanical properties from their counterparts growing in windy sites [14]; and tree form is also influenced by the amount of competition for growing space [...] [15].”

This is the reason why we think that the so-called “mechanics” in the WBE and WBE-related models needs to be revisited and discussed more thoroughly. And we prefer to speak about “elastic-similarity” rather than “mechanics” when dealing with WBE and WBE-related models.

Second, we think that the statement made in the introduction that in the WBE model, “different allometric scaling laws can be derived without explicit reference to hydraulics” is not *wrong*. Indeed we wrote an entire section in the Supplementary Discussion (section 1.3) showing mathematically how allometric laws can be derived from a set of hypotheses of WBE without any use of hydraulics. These hypotheses are clearly mentioned: (i) Self-similarity, (ii) Volume filling ($D=3$), (iii) Elastic similarity. In fact, all the allometric laws used in our paper (height vs. trunk diameter, leaf biomass vs. trunk diameter, and leaf biomass vs. stem biomass, see Fig. 6), can be recovered with these three hypotheses and do not make necessary the use of hydraulics in WBE. Therefore, if the reviewer still believes that our manuscript is “confusing and wrong at several levels”, we would expect him/her to be more specific about how and where our demonstration can be falsified.

That been said, we agree with the reviewer: “many predictions of WBE require both the mechanics and hydraulics”. Indeed, hydraulics is important, for example to derive allometric relations for fluid velocity, pressure gradient, etc., as it is mentioned in the introduction (we have slightly rephrased this part):

“With these hypotheses [self-similarity, volume filling, and elastic similarity], several allometric relations between trunk radius, tree height, stem biomass, and leaf biomass can be deduced without explicit reference to hydraulics. Additionally, considering the minimisation of hydraulic losses within this mechanically-based structure, the WBE model also shows that fluid velocity, hydraulic conductance, pressure gradient, and tube diameter vary allometrically with plant mass [20].”

To clarify our point of view and address the reviewer’s comment, we also revised two sentences in the Discussion:

“The WBE model is an analytical model based on geometrical, mechanical, and hydraulic arguments. The geometrical and mechanical hypotheses of WBE are (i) self-similarity, (ii) volume filling ($D = 3$), and (iii) elastic similarity. These hypotheses are analogous to the three hypotheses of the proposed AMT model, except that the major mechanical argument is elastic buckling instead of wind-induced pruning. The allometric relations that can be derived from these hypotheses are very similar to the predictions of the AMT model (Table III, Fig. 6, Supplementary discussion, section 1.3). It is thus impossible to refute either approach based on allometry only.”

“We believe that further insights could be gathered by combining the biomechanically-based model of plant-wind interactions proposed in the present paper with the hydraulic hypotheses of the WBE model to have a better understanding of both mechanics and hydraulics.”

Finally, the reviewer claims that “the WBE model in any form predicts self-similarity but does not assume self-similarity”. Again, we cannot agree with that. Among the hypotheses of the WBE are self-similarity, volume filling, and elastic similarity (or resistance to buckling).

“The model is based on a few general principles: (1) the branching network is volume filling; (2) the leaf and petiole size are invariant; (3) biomechanical constraints are uniform; and (4) energy dissipated in fluid flow is minimized.” — G. B. West, J. H. Brown, B. J. Enquist. 1999. A general model for the structure and allometry of plant vascular systems. Nature, 400, 66-667.

*“The design of trunks and branches to resist buckling leads to some optimal relationship between their length and radius: $l_k \sim r_k^\alpha$.” — *ibid*.*

“Analyses based on solutions to the bending moment equations for beams give $\alpha = 2/3$ (refs 2, 20).” — *ibid.*

Therefore, self-similarity is indeed one of the hypotheses of the WBE theory, not a prediction. This contrast with MechaTree, for which self-similarity is an emergent property, as is it mentioned in the Discussion:

“The existence of branching, length, and diameter ratios shows that tree architectures are self-similar (more rigorously, they are self-affine since the different lengths can vary differently with scale). Yet, the whole structure is an assembly of segments of same length L. Self-similarity is thus an emergent property resulting from the complex architecture reorganisations that occur during the life-time of a tree through wind-induced pruning and light- and wind-dependent growth.”

d. The authors keep talking about hydraulic models when in fact many of these models are a combination of hydraulics and mechanics. That is, they are not purely hydraulic, and this need to be stated more clearly and explicitly when the term hydraulic is used by itself so much. The authors seem to not acknowledge that good models already exist that incorporate hydraulics and mechanics and have predictions that rely on both and yield different types of predictions.

We agree with the reviewer that our description of the hydraulic and mechanical aspects was not accurate enough sometimes. During the revision, we have thus tried to clarify these aspects throughout the text (including the abstract), in particular wind biomechanics vs. buckling under self-weight and elastic similarity. We now hope that the difference between the different models, MechaTree, AMT and WBE, appears more clearly.

QUESTION 3

There are a few typos such as this missing "be": "The dependence of the fractal dimension on the foliage transparency can BE interpreted as follows." The manuscripts should be proofread carefully once more.

We thank the reviewer for pointing out this typo that has been corrected. We have also carefully proofread our manuscript once more. As always, there may be a few typos remaining though, but we trust the editorial office of Nature Communications to handle this matter.

REFERENCES

- A. Albrecht, E. Badel, V. Bonnesoeur, Y. Brunet, T. Constant, P. Défossez, E. de Langre, S. Dupont, M. Fournier, B. Gardiner, S. J. Mitchell, J. R. Moore, B. Moulia, B. C. Nicoll, K. J. Niklas, M. J. Schelhaas, H. C. Spatz, F. W. Telewski. Comment on “Critical wind speed at which trees break”. *Phys. Rev. E* 94, 067001, 2016.
- E. de Langre. Effects of wind on plants, *Annu. Rev. Fluid Mech.* 40, 141–168, 2008.
- R. A. Duursma, D. S. Falster, F. Valladares, F. J. Sterck, R. W. Pearcy, C. H. Lusk, K. M. Sendall, M. Nordenstahl, N. C. Houter, B. J. Atwell, et al. Light interception efficiency explained by two simple variables: a test using a diversity of small-to medium-sized woody plants. *New Phytol.* 193, 397–408, 2012.
- B. Gardiner, P. Berry, and B. Moulia. Review: Wind impacts on plant growth, mechanics and damage. *Plant Sci.*, 245:94–118, 2016.
- C. Godin and H. Sinoquet. Functional–structural plant modelling. *New Phytol.*, 166:705–708, 2005.

- T. A. McMahon. Size and shape in biology. *Science*, 179, 1201, 1973.
- L. Poorter, R. Kwant, R. Hernandez, E. Medina, and M. J. A. Werger. Leaf optical properties in Venezuelan cloud forest trees. *Tree Physiol.*, 20:519–526, 2000.
- G. B. West, J. H. Brown, B. J. Enquist. A general model for the structure and allometry of plant vascular systems. *Nature*, 400, 66-667, 1999.

Reviewers' comments:

Reviewer #2 (Remarks to the Author):

I am a big fan of this paper, and I am very pleased to see that a sensitivity analysis has now been conducted and included!

However, I still think the authors are confused on point 2 from my last review. One main reason I continue to discuss this is because I do not want the reception of this excellent paper to be harmed by negative reactions to misunderstandings or misrepresentations of previous models. I also just want to see things accurately reported.

I now specifically respond to the specific comments of the authors. The authors do not have a correct understanding of what are the assumptions or predictions in the WBE model. My best advice to them at this point is to compare results with the WBE model (and similar models would be nice too) but to minimize or be very safe or perhaps completely eliminate their discussion about the assumptions and predictions of the WBE model. If the authors do want to focus on what are assumptions and predictions in the WBE model, I hope they carefully read the detailed discussion below

1. The most fundamental confusion seems to be when the authors say:
"Finally, the reviewer claims that "the WBE model in any form predicts self-similarity but does not assume self similarity". Again, we cannot agree with that. Among the hypotheses of WBE are:

a. self-similarity, b. volume filling, and c. elastic similarity (or resistance to buckling).

'The model is based on a few general principles:

- (1) the branching network is volume filling;
- (2) the leaf and petiole size are invariant;
- (3) biomechanical constraints are uniform; and
- (4) energy dissipated in fluid flow is minimized.'

— G. B. West, J. H. Brown, B. J. Enquist. 1999. A general model for the structure and allometry of plant vascular systems. *Nature*, 400, 66-667.

I find the above logic very confusing. The authors state assumptions a-c as WBE and then give a quote from WBE with their key assumption 1-4. So let's be very explicit about this. Author's stated WBE assumption b maps to quoted WBE assumption 1 for volume filling. Author's stated WBE assumption c maps to quoted WBE assumption 3 for elastic similarity. Author's stated WBE assumption a for self similarity, which is the main point of contention, does not map to any of the quoted WBE assumptions 1-4. In reverse, WBE quoted assumption 2 about invariant terminal units and assumption 4 about hydraulics are not included in any of the author's stated WBE assumptions, which again gets back to the idea of not fully including hydraulics in their discussion.

To clarify, further the quoted WBE assumptions 1-4 are exactly right and are what are used to derive and predict self similarity. That is, self similarity is a consequence of assumptions 1-4, but it is not contained anywhere in assumptions 1-4. WBE never assume it, and that is why it is absent from the quoted 1-4 list. Perhaps the main contribution of WBE is that it derive self similarity without assuming it. That is, volume filling can be used to derive that $\Gamma = l_{k+1}/l_k = n^{-1/3} = \text{constant}$ which is the mathematical statement that the ratio of length from one level to the next is invariant, and that is the statement of self similarity and that is the end result that follows from the assumption of volume filling. Self similarity is not assumed. Volume filling is assumed and that lead to a prediction of self similarity for lengths.

To clarify yet further, this again is just one sets of assumptions for the original WBE model. There are many subsequent models like this that used modified or additional assumptions and for all of those, self similarity is derived and predicted, not assumed.

To respond to the other quote supplied by the authors, that "The design of trunks and branches to resist buckling leads to some optimal relationship between their length and radius: $l_k \sim r_k^\alpha$.' — ibid." This is correct. Math is done to find an optimal solution that satisfies the quoted assumption 1-4 listed above, and the result is this equation for the optimal relationship that implies self similarity, and this means that result (self similarity) is the derived or predicted consequence of the quoted base assumptions 1-4.

2. The authors also claim "Second, we think that the statement made in the introduction that in the WBE model, "different allometric scaling laws can be derived without explicit reference to hydraulics" is not wrong. Indeed we wrote an entire section in the Supplementary Discussion (section 1.3) showing mathematically how allometric laws can be derived from a set of hypotheses of WBE without any use of hydraulics. These hypotheses are clearly mentioned: (i) Self-similarity, (ii) Volume filling ($D=3$), (iii) Elastic similarity."

As just explained in point 1, WBE does NOT assume self similarity and therein lies the crux of the disagreement on this point. The self similarity arises from the quoted WBE assumptions 1-4, which includes hydraulics (i.e., assumption 4), and that is why within WBE and similar models, hydraulics are essential and necessary to even get to the point from which the authors begin.

3. The authors say, "This is the reason why we think that the so-called mechanics in the WBE and WBE-related models needs to be revisited and discussed more thoroughly. And we prefer to speak about elastic-similarity rather than mechanics when dealing with WBE and WBE-related models." My main point here is that there is a big difference between questioning the mechanics included in a model and saying it does NOT include mechanics. Elastic-similarity is a type of mechanics, and if the authors do not like it, they should challenge by saying it is the wrong type of mechanics to include, which they argue quite convincingly in their response letter, but they should not say it is NOT a type of mechanics or is "so-called" mechanics.

RESPONSE TO REVIEWER #2

I am a big fan of this paper, and I am very pleased to see that a sensitivity analysis has now been conducted and included!

However, I still think the authors are confused on point 2 from my last review. One main reason I continue to discuss this is because I do not want the reception of this excellent paper to be harmed by negative reactions to misunderstandings or misrepresentations of previous models. I also just want to see things accurately reported.

Thank you again. We appreciate this favourable evaluation of our work.

Like the reviewer, we hope that there will be a good reception of our work. During the revision, we have really tried to avoid counterproductive disputes, especially those involving presentations of previous models. We should add that our aim has never been to rule out or discredit previous models, but to provide an alternative and novel approach to tree allometry.

I now specifically respond to the specific comments of the authors. The authors do not have a correct understanding of what are the assumptions or predictions in the WBE model. My best advice to them at this point is to compare results with the WBE model (and similar models would be nice too) but to minimize or be very safe or perhaps completely eliminate their discussion about the assumptions and predictions of the WBE model. If the authors do want to focus on what are assumptions and predictions in the WBE model, I hope they carefully read the detailed discussion below

In revising the manuscript, we have followed the reviewer's suggestion: we have minimized the discussion on the assumptions and predictions of the WBE model. In particular, we felt that part of the disagreement came from the following two sentences in the previous version of the manuscript:

"With these hypotheses, several allometric relations between trunk radius, tree height, stem biomass, and leaf biomass can be deduced without explicit reference to hydraulics."

"The geometrical and mechanical hypotheses of WBE are (i) self-similarity, (ii) volume filling ($D = 3$), and (iii) elastic similarity."

The first sentence has been changed into:

"With this approach, several allometric laws can be deduced, relating trunk radius, tree height, stem biomass, and leaf biomass."

And the second sentence has been removed. We hope that the reviewer will now feel that the manuscript is suitable for publication. Please, find a point-by-point answer below. Changes in the manuscript and in the Supplementary Information have been coloured in red for convenience.

QUESTION 1

The most fundamental confusion seems to be when the authors say:

"Finally, the reviewer claims that "the WBE model in any form predicts self-similarity but does not assume self similarity". Again, we cannot agree with that. Among the hypotheses of WBE are:

a. self-similarity, b. volume filling, and c. elastic similarity (or resistance to buckling).

'The model is based on a few general principles: (1) the branching network is volume filling; (2) the leaf and petiole size are invariant; (3) biomechanical constraints are uniform; and (4) energy dissipated in fluid flow is minimized.' — G. B. West, J. H. Brown, B. J. Enquist. 1999. A general model for the structure and allometry of plant vascular systems. *Nature*, 400, 66-667.

I find the above logic very confusing. The authors state assumptions a-c as WBE and then give a quote from WBE with their key assumption 1-4. So let's be very explicit about this. Author's stated WBE assumption b maps to quoted WBE assumption 1 for volume filling. Author's stated WBE assumption c maps to quoted WBE assumption 3 for elastic similarity. Author's stated WBE assumption a for self similarity, which is the main point of contention, does not map to any of the quoted WBE assumptions 1-4. In reverse, WBE quoted assumption 2 about invariant terminal units and assumption 4 about hydraulics are not included in any of the author's stated WBE assumptions, which again gets back to the idea of not fully including hydraulics in their discussion.

To clarify, further the quoted WBE assumptions 1-4 are exactly right and are what are used to derive and predict self similarity. That is, self similarity is a consequence of assumptions 1-4, but it is not contained anywhere in assumptions 1-4. WBE never assume it, and that is why it is absent from the quoted 1-4 list. Perhaps the main contribution of WBE is that it derive self similarity without assuming it. That is, volume filling can be used to derive that $\Gamma = l_{k+1}/l_k = n^{-1/3} = \text{constant}$ which is the mathematical statement that the ratio of length from one level to the next is invariant, and that is the statement of self similarity and that is the end result that follows from the assumption of volume filling. Self similarity is not assumed. Volume filling is assumed and that lead to a prediction of self similarity for lengths.

To clarify yet further, this again is just one sets of assumptions for the original WBE model. There are many subsequent models like this that used modified or additional assumptions and for all of those, self similarity is derived and predicted, not assumed.

Part of the argument seems to come from different definitions of self-similarity. To clarify our point of view, and to avoid any ambiguity, we now state the first hypothesis of the AMT model as “self-similarity of the tree skeleton”, instead of simply “self-similarity”. By this we mean (section “Self-similar ratios”):

“The existence of branching and length ratios, R_n and R_l , both independent of k , means that the tree skeleton is self-similar and that a fractal dimension can be defined: $D = \ln R_n / \ln R_l$.”

Initially, we believed that the WBE model make implicitly use of a similar hypothesis (namely “self-similarity of the tree skeleton” or self-similarity of branch lengths).

Now, since we agree with the reviewer that we should avoid any source of confusion on that topic, we removed any mention of the assumptions of the WBE models in the main text. And in the Supplementary Discussion, we now introduce carefully our personal derivation of the allometric scalings of the WBE model by using the word “principle” instead of “assumption” and by specifying that the principle of the self-similarity is reduced to the “skeleton” of the tree

“The WBE model should be compared with the AMT model, as only comparisons between models of similar types (in this case, static optimal models) are meaningful. To identify the similarities and differences between these two models, it is useful to derive them from similar principles, using the same notations. Note however that alternative derivations of the WBE models are possible, based on different assumptions. Here, we derive the WBE equations from three principles:

i. *Self-similarity of the tree skeleton;*

ii. Volume filling;

iii. Pipe model or elastic similarity.”

To respond to the other quote supplied by the authors, that "The design of trunks and branches to resist buckling leads to some optimal relationship between their length and radius: $l_k \sim r_k^\alpha$," — *ibid.*" This is correct. Math is done to find an optimal solution that satisfies the quoted assumption 1-4 listed above, and the result is this equation for the optimal relationship that implies self similarity, and this means that result (self similarity) is the derived or predicted consequence of the quoted base assumptions 1-4.

We agree with the reviewer, there is indeed a possibility to derive the elastic similarity scaling law ($l_k \sim r_k^{2/3}$) from the pipe model. This alternative is the method that what used in the initial version of the WBE model (1997). This is now clarified in the Supplementary Discussion (section 1.3):

“For the third principle one has a choice: as already noted by Savage et al. [4], both the area conservation associated with the pipe model [6] (used in the original WBE model [1]) and the elastic similarity scaling law (used in later versions of the WBE model [2]) yield similar relationships when combined with volume filling. As this work is centered on plant biomechanics we retain elastic similarity here.”

In the 1999 version of the WBE model, the one we use as a comparison with our AMT model, the pipe model is not used, and we believe the hypothesis of elastic similarity is used instead. Indeed the authors of the paper state:

*“The design of trunks and branches to resist buckling leads to some optimal relationship between their length and radius: $l_k \sim r_k^a$. Comparison with equation (1) in Box 1 gives $a = 2 / (3\alpha)$. If this holds uniformly for all branches, then a is constant, independent of k , in which case a and $\beta_k = r_{k+1} / r_k$ are also constant. When coupled with the volume-filling constraint, $\gamma_k = n^{-1/3}$, this proves that the branching architecture is a self-similar fractal^{18,19}. Analyses based on solutions to the bending moment equations for beams give $\alpha = 2/3$ (refs 2, 20). This is most important for the trunk and large branches^{7,15,20}. Assuming that it holds for all branches (all k) leads to $a = 1$, which is the condition for area-preserving branching³. **Unlike a previous report³, this derivation does not assume the pipe model.**” — G. B. West, J. H. Brown, B. J. Enquist. 1999. A general model for the structure and allometry of plant vascular systems. *Nature*, 400, 66-667.*

To add to this argument, the authors clearly mention that the tapering of branches (which is related to the a exponent, whereas the tapering of hydraulic conduits is related to \bar{a}) can be estimated from mechanical (not hydraulic) considerations:

*“Box 2 shows how \bar{a} , and consequently the degree of tapering, are determined from hydrodynamic considerations, whereas a is determined from mechanical constraints.” — *ibid.**

QUESTION 2

The authors also claim "Second, we think that the statement made in the introduction that in the WBE model, "different allometric scaling laws can be derived without explicit reference to hydraulics" is not wrong. Indeed we wrote an entire section in the Supplementary Discussion (section 1.3) showing mathematically how allometric laws can be derived from a set of hypotheses of WBE without any use of hydraulics. These hypotheses are clearly mentioned: (i) Self-similarity, (ii) Volume filling ($D=3$), (iii) Elastic similarity."

As just explained in point 1, WBE does NOT assume self similarity and therein lies the crux of the disagreement on this point. The self similarity arises from the quoted WBE assumptions 1-4, which includes hydraulics (i.e., assumption 4), and that is why within WBE and similar models, hydraulics are essential and necessary to even get to the point from which the authors begin.

As stated before, we believe both points can be reconciled. Some of the WBE allometric scalings can be derived without explicit reference to hydraulics. That does not mean that it is the orthodox derivation, or that it is the only derivation possible.

However, this point does not concern the manuscript anymore since we have removed all references to WBE hypotheses, following the reviewer's suggestion. We thus believe that it is not necessary to pursue the discussion on the subject.

QUESTION 3

The authors say, "This is the reason why we think that the so-called mechanics in the WBE and WBE-related models needs to be revisited and discussed more thoroughly. And we prefer to speak about elastic-similarity rather than mechanics when dealing with WBE and WBE-related models." My main point here is that there is a big difference between questioning the mechanics included in a model and saying it does NOT include mechanics. Elastic-similarity is a type of mechanics, and if the authors do not like it, they should challenge by saying it is the wrong type of mechanics to include, which they argue quite convincingly in their response letter, but they should not say it is NOT a type of mechanics or is "so-called" mechanics.

We agree with the reviewer: our previous statement about elastic similarity was certainly exaggerated. However, this statement does not appear in the manuscript. It was only intended to fuel the exchange with the reviewer and the editor in our previous response. Hopefully, this dispute can now be closed.

REVIEWERS' COMMENTS:

Reviewer #2 (Remarks to the Author):

This main focus of this paper is well reasoned and the figures/videos and results are beautiful. The revision goes a very long way towards addressing my concerns. However, there is still a statement and paragraph in the supplementary discussion about the WBE model that is just not correct. The statement is that the authors say they are comparing with the WBE model, and then they change the assumptions/principles of the WBE model. The WBE model is defined by its assumptions/principles more than its results, so to change the assumptions/principles is to change the model and make it not be the WBE model. As stated now in several rounds of review, the WBE model does not assume self similarity. Self similarity is a consequence. This discord is apparent in the authors own writing here. They say the WBE model is a hydraulics model but then none of the principles they list for the model involve hydraulics. This would seem to be a contradiction, but the resolution is that the first assumption/principle should be minimization of power of water/sap flow through the plant and that leads to self similarity. That is, the first assumption/principle is indeed about hydraulics, and self similarity is a consequence of that. The problematic paragraph is "Principle (i) is shared by the WBE and the AMT models: tree skeletons are assumed to be self-similar structures. Principle (ii) is based on an analogy with arterial and bronchial networks, for which volume-filling makes sense because every part of the volume should be irrigated by arteries or bronchia. Principle (iii) states that branches should follow the scaling law $l_k / d_k^2 = 3k$, such that the deflection of their tip $\propto k$ under self-weight is proportional to l_k for a given orientation [5]. The same scaling is found for upright branches to provide an optimal resistance to elastic buckling [2–4]." There are two problems here. First, as just explained, self similarity is simply not a shared assumption of the AMT and WBE model because it is not an assumption of the WBE model. Second, Principle (iii) does not state that anything should follow a power law. The power law is a consequence of the assumption/principle. The assumption/principle is a statement, and after additional derivation and work, one derive the power law and show it is a consequence.

RESPONSE TO REVIEWER #2

This main focus of this paper is well reasoned and the figures/videos and results are beautiful. The revision goes a very long way towards addressing my concerns.

We thank the reviewer again for this positive evaluation of our manuscript.

However, there is still a statement and paragraph in the supplementary discussion about the WBE model that is just not correct. The statement is that the authors say they are comparing with the WBE model, and then they change the assumptions/principles of the WBE model. The WBE model is defined by its assumptions/principles more than its results, so to change the assumptions/principles is to change the model and make it not be the WBE model. As stated now in several rounds of review, the WBE model does not assume self similarity. Self similarity is a consequence. This discord is apparent in the authors own writing here. They say the WBE model is a hydraulics model but then none of the principles they list for the model involve hydraulics. This would seem to be a contradiction, but the resolution is that the first assumption/principle should be minimization of power of water/sap flow through the plant and that leads to self similarity. That is, the first assumption/principle is indeed about hydraulics, and self similarity is a consequence of that. The problematic paragraph is "Principle (i) is shared by the WBE and the AMT models: tree skeletons are assumed to be self-similar structures. Principle (ii) is based on an analogy with arterial and bronchial networks, for which volume-filling makes sense because every part of the volume should be irrigated by arteries or bronchia. Principle (iii) states that branches should follow the scaling law $l_k / d_k = 3k$, such that the deflection of their tip k under self-weight is proportional to l_k for a given orientation [5]. The same scaling is found for upright branches to provide an optimal resistance to elastic buckling [2–4]." There are two problems here. First, as just explained, self similarity is simply not a shared assumption of the AMT and WBE model because it is not an assumption of the WBE model. Second, Principle (iii) does not state that anything should follow a power law. The power law is a consequence of the assumption/principle. The assumption/principle is a statement, and after additional derivation and work, one derive the power law and show it is a consequence.

In response to the reviewer's comment, we have revised the "problematic paragraph" of the Supplementary Discussion. The three assumptions/principles now refer to "the present derivation of the WBE results" instead of the "WBE model" to avoid any ambiguity. Beside the "principles" used in the seminal papers have been clearly differentiated from the "assumptions" of our proposed derivation. Finally, the description of assumption (iii) has been rewritten as suggested by the reviewer (i.e. first the biophysical principle, then its consequence on power law scalings). Changes are highlighted in red for convenience. We really believe that this revision will permit to avoid any confusion that the reviewer feared.

"The WBE model should be compared with the AMT model, as only comparisons between models of similar types (in this case, static optimal models) are meaningful. To identify the similarities and differences between these two models, it is useful to derive their results from similar assumptions, with the same notations. Note that the present derivation of WBE allometric equations does not use the first principles outlined in the seminal WBE papers [1, 2]. Note also that alternative derivations of the WBE equations are possible, based on different assumptions. Here, we derive the WBE allometric scalings from three assumptions:

- i. Self-similarity of the tree skeleton;*
- ii. Volume filling;*
- iii. Pipe model or elastic similarity.*

For the third assumption one has a choice: as already noted by Savage et al. [4], both the area conservation associated with the pipe model [6] (used in the original WBE model [1]) and the elastic similarity scaling law (used in later versions of the WBE model [2]) yield similar relationships when combined with volume filling. As this work is centered on plant biomechanics we retain elastic similarity here.

Assumption (i) is shared the AMT model and the present derivation of the WBE results: tree skeletons are assumed to be self-similar structures (here self similarity is assumed for simplicity; it was found to emerge in the MECHATREE numerical model, or was derived from the principles of minimal hydraulic conductance and volume filling in the seminal derivation of the WBE model for plants [2]). Assumption (ii) is based on an analogy with arterial and bronchial networks, for which volume-filling makes sense because every part of the volume should be irrigated by arteries or bronchia. Assumption (iii), or elastic similarity, states that aspect ratio of branches has been selected such that the deflection of their tip Δ_k under self-weight is proportional to their length l_k for a given orientation [5], and, for upright branches, to provide an optimal resistance to elastic buckling [2–4]. It was found that these branches should follow the scaling law $l_k \propto d_k^{2/3}$.